# A Review of Technologies and Techniques for Indoor Navigation Systems for the Visually Impaired

**DOI:** 10.3390/s20143935

**Published:** 2020-07-15

**Authors:** Walter C. S. S. Simões, Guido S. Machado, André M. A. Sales, Mateus M. de Lucena, Nasser Jazdi, Vicente F. de Lucena

**Affiliations:** 1PPGI/ICOMP—Programa de Pós-Graduação em Informática, Institute of Computing, UFAM—Federal University of Amazonas, Manaus, AM 69080-900, Brazil; vicente@ufam.edu.br; 2PPGEE—Programa de Pós-Graduação em Engenharia, Technology College, UFAM—Federal University of Amazonas, Manaus, AM 69080-900, Brazil; guidosopranomachado@gmail.com (G.S.M.); andrealgaranhaes@gmail.com (A.M.A.S.); 3Software/Hardware Integration Lab, UFSC—Federal University of Santa Catarina, Florianópolis, SC 88040-900, Brazil; lucena@lisha.ufsc.br; 4Institute of Industrial Automation and Software Systems, The University of Stuttgart, 70550 Stuttgart, Germany; Nasser.jazdi@ias.uni-stuttgart.de; 5CETELI–Sector North of UFAM’s Main Campus, UFAM—Federal University of Amazonas, Manaus, AM 69080-900, Brazil

**Keywords:** indoor positioning system, location techniques, data fusion, comparisons of proposals, indoor mapping

## Abstract

Technologies and techniques of location and navigation are advancing, allowing greater precision in locating people in complex and challenging conditions. These advances have attracted growing interest from the scientific community in using indoor positioning systems (IPSs) with a higher degree of precision and fast delivery time, for groups of people such as the visually impaired, to some extent improving their quality of life. Much research brings together various works that deal with the physical and logical approaches of IPSs to give the reader a more general view of the models. These surveys, however, need to be continuously revisited to update the literature on the features described. This paper presents an expansion of the range of technologies and methodologies for assisting the visually impaired in previous works, providing readers and researchers with a more recent version of what was done and the advantages and disadvantages of each approach to guide reviews and discussions about these topics. Finally, we discuss a series of considerations and future trends for the construction of indoor navigation and location systems for the visually impaired.

## 1. Introduction

The estimation of a target’s position in an outdoor environment is generally resolved using a Global Satellite Navigation System (GNSS) [1]. However, in an indoor environment, we still need to define a positioning system standard that can be applied worldwide and in different contexts. For this reason, the research efforts of authors working in the field of positioning have recently focused on scenarios. An indoor positioning system (IPS) is designed to provide information about a person’s position within a building. The evolution of IPS has facilitated the creation of indoor location-based services (ILBSs), which create applications based on position knowledge. Examples of this type of service are locating people in a shopping mall, tracking patients inside a hospital, advising firefighters inside a building, advising people inside an airport, or developing assisted living systems for the elderly and visually impaired. Therefore, there is great interest in developing IPSs that can be applied in all construction units, regardless of their physical structure. In general, an IPS has three requirements to be met for large-scale use: (i) the system must provide accurate position estimates, (ii) the system must be easily scalable, and (iii) the cost of implementation must be reduced.

The current trend to reduce system cost is to use the wireless infrastructure already deployed for communication as reference points for positioning [2]. Many communication technologies are available for use in indoor environments, among them are, LTE, Wi-Fi, Bluetooth, Bluetooth Low-Energy (BLE), wireless sensor networks (WSNs), and ultra-wide band (UWB). The most used so far is Wi-Fi technology. In addition to Wi-Fi network devices, other technologies have been investigated to build more robust schemes and bring the level of indoor accuracy closer to the results already perceived with outdoor systems. Among the technologies are inertial, sound, light, and visual systems.

Micro-electromechanical systems (MEMS) provide low-cost inertial sensors that can estimate the positions of pedestrians without the need to implement any infrastructure in building units [3]. Sound systems use hardware that emits audible and inaudible sounds and make it possible to insert coding that provides information to the system without disturbing human users [4]. Light-based systems use natural and artificial light to define points of interest in location systems. In these light signals, it is also possible to insert coding that acts as a language, containing instructions for the localization system [5].

Vision-based systems use cameras with various characteristics and can be applied in two ways, in the infrastructure or on a mobile device [6]. When applied to the infrastructure, the cameras are arranged to cover the largest visual field so that the targets can be located and tracked. When cameras are used on mobile devices, the objective is to cover the largest visible area around the user and visually identify information registered as a reference.

Most of these technologies are already available on today’s smartphones, which allows researchers to focus their efforts on defining algorithms that manipulate signals and data obtained from sensors to provide more reliable information.

Several studies have attempted to review the current state-of-the-art of IPS. Since the first review, several advances have been made in these studies, showing that the area is still open and there are several gaps to be corrected or filled. For example, Alakhras et al. described the use of ubiquitous technologies for indoor location systems, and Kuang et al. advanced the study, introducing new technical and technological approaches [7,8]. From this work, the latest reviews were carried out, mainly advancing hybrid approaches [9,10,11]. Still, within indoor positioning system studies, other research has been directed toward the construction of maps, which allow relating the records to provide routes to users, as in the investigations of Teng et al., Zheng et al., and Jung et al. [12,13,14].

Despite the numerous, quite comprehensive, studies of the state-of-the-art of IPSs, it is always necessary to revisit the evolution of technologies and techniques and to make advances in the domain of resources to make the systems of location, navigation, and tracking in indoor environments more robust. In this work, we present an overview of the state-of-the-art of IPSs, analyzing the advantages and disadvantages of the systems described by the authors of the referenced works, with a focus on providing a higher level of accuracy in order to serve visually impaired users. First, we classify and review individual IPSs, and then we highlight hybrid systems.

The rest of the paper is organized as follows: Section 2 presents a set of surveys that deal with technological and technical updates of the resources used in indoor positioning systems. Section 3 examines radio-based, inertial, sound-based, light-based, and computer-based systems. Hybrid systems are also covered in this section, indicating various technological and technical combinations (algorithms). Section 4 presents a comparison of the studies discussed in this review, showing the advantages and disadvantages of each approach. Finally, in Section 5, we present the conclusions and some possible future lines of work.

## 2. Related Work

The research area dealing with indoor positioning systems has received many contributions in recent years. Although many products are already available for use, this is a topic still considered open for discussion. Even the most modern approaches need updates due to the evolution of hardware and interventions in the mathematical models that define the construction of the algorithms, as widely described by authors such as Wang et al., Yassin et al., Khelifi et al., Wu et al., Mainetti et al., and Zafari et al. [10,15,16,17,18,19].

Another issue described in several studies is the need to establish clear objectives, which can be achieved by experimentation and allow replication, attracting more interest from those who seek to develop criteria for the construction and evaluation of projects. The requirements identified as feasible to be met and proven are precision, speed, and a balance between the two factors [20]. For example, Nguyen et al. described the concern to find a balance between maintaining a high level of accuracy without affecting the delivery time of information to visually impaired users when using internal navigation systems [21].

The list of works selected for the construction of this research mainly covers those that discriminate their technological and technical approaches. The works cited in this paper are concentrated in the period 2011 to 2020. Particular attention is paid to the period after 2015. Our goal is to highlight a more current view of where the advances in the area are located. Two investigation profiles allow the construction of this review: review-type works and applied works (journals, papers). Review-type works indicate research organization strategies that other authors have prepared, and other works (excluding reviews) are used to show the results achieved in practice using the technological and technical approaches presented in the reports. This research presents the strategies that use specific technologies and methods that combine different hardware and algorithms to compose hybrid arrangements, which seek greater precision robustness, error control, and response time, as described by Yassin and Gharghan [16,22].

The subject is quite comprehensive when discussing the composition of models, systems, and objectives [23]. Although the use of such systems allows the location, tracking, and navigation of targets in internal environments, their construction processes are established to meet precision and punctuality requirements defined jointly by the users responsible for the development of the system and the limits set by the hardware and software layers [24].

In general, fast applications use simple algorithms that ignore variables and some processes and require less processing and memory usage [25,26]. More complex applications provide more complete data, and therefore more accurate systems; however, they need more time to perform their calculations [2,27]. Applying more simplified algorithm formulations linearizes the problems and can deliver results more quickly, within tolerable limits of variation of results, as indicated by Wang and Chow [15,28]. Complex algorithms are justified by the deepening of knowledge about the data, requiring more time to separate essential data from data that contribute little or indicate redundancy of information, as reported by Mohamed and Chen [29,30]. Some authors use hybrid approaches that find a balance between the two variables [16,31]. Knowing the classes of logical formulations, as well as associated techniques and technologies, is an essential requirement for researchers to define their IPS projects.

Some authors establish criteria for organizing their surveys to describe the standards adopted in indoor location and navigation projects. In these criteria, sometimes authors choose to group techniques and technologies, and sometimes they go into more detail, always focusing on the final reader and accelerating the understanding of the resources available for their research.

It is possible to organize physical IPS structures in different ways, according to the author’s interest. Noticeably, many of these ways end up leading to a more simplified organization, which only has three approaches: those that require the construction or use of external resources, organized in the environment to meet specific demands, those that are contained on a mobile device, and approaches that unite more than one proposal, called mixed or hybrid [32,33,34]. We can also organize logical structures into three approaches: linear, nonlinear, and hybrid [35].

Yassin et al. wrote a survey to indicate recent advances in IPS approaches in using technologies based on radio signals and Wi-Fi networks [16]. The authors adopted a more detailed approach so that the reader could understand the methods used in the papers cited in the work. There is a grouping scheme for measurement techniques (lateration, angulation, fingerprint, Cell-ID, radio frequency identification (RFID)), and another scheme to explain the algorithms (triangulation, scene analysis, and proximity detection), accompanied by application sites and level of accuracy. Yen et al. extended their studies to approaches that approach methods based on the k nearest neighbor (KNN) and methods of the weighted k nearest neighbor (WKNN) [36]. The WKNN method was designed to adjust the weights for each of the K coordinates based on their corresponding errors in the relationship, where smaller errors introduce larger weights.

Khelifi et al. also researched the use of radio- and Wi-Fi-based technologies [17]. The authors extended the discussion of the topic beyond precision, indicating that attention is needed on issues such as processing cost, the monetary cost of implementation, and energy efficiency [17]. The related works used to support the study were grouped in terms of techniques and technologies, as this makes reading more objective and improves understanding by the reader.

Wu et al. extended the discussion on the use of technologies and techniques to locate people indoors [10]. The authors experimented with the addition of inertial sensors, such as magnetometer, gyroscope, and accelerometer, to get direction indications with more reliability. The built system uses the inertial sensor-based pedestrian dead-reckoning (IPDR) algorithm to navigate people in indoor environments safely. The system updates the values that are above the error limits with the Kalman filter, which increases the robustness of the final system information.

Mainetti et al. sought to cover the main IPS technologies and techniques used to locate, map, track, and navigate people, objects, or animals indoors [18]. The authors describe computer vision (mobile and fixed cameras), infrared, ultrasound, Wi-Fi, RFID, and Bluetooth as the main technologies used in IPS. The main techniques are described in detail, and the others are grouped into a class called “other technologies,” as is the case with approaches that use ZigBee and FM radio technology. The authors followed a model to present the results of the works used as references in graph and table format, using the criteria of precision, coverage, cost, complexity, and typical implementation locations.

Zafari et al. focused on the factors of energy efficiency, signal latency, and scalability of indoor location systems when used on mobile devices [19]. The authors organized the work literature into three main groups: device-based location (DBL), monitor-based location (MBL), and proximity detection. The work shows a comparison of the gains and losses of algorithms and devices used in the related works, considering the difficulties of operation in real environments, such as signal losses, scenario interference in the sensors, and noise from one sensor to another.

Davidson et al. established as a criterion for the construction of an indoor location survey the use of sensor grouping in schemes called clusters [37]. This survey was restricted to sensors available on smartphones and problems such as direction indication and floor localization. An exciting part of the review indicates open questions, such as how to include autonomous resources in the system, integrate external monitoring systems, define maintenance policies, update maps, and continuously improve the performance of indoor positioning systems.

Singh et al., in their study, highlighted the difficulty of identifying addresses mapped using single sensor information or the use of a unique technique [38]. The research was directed to the combination of records established by different technologies in clusters to increase the representativeness of the records, which can assume static or dynamic characteristics. A concern of the authors in determining the arrangements is to balance the number of records activated to identify an address and the energy consumption spent by the settlement, even when observed in critical situations such as connection failures or the significant presence of noise.

Plikynas et al. indicated that there are currently many solutions available to assist in the navigation of people with visual impairments in previously known indoor environments [24]. The authors suggested that indoor environments have more significant complexities than outdoor environments, and alerts should be given more quickly to avoid accidents. In reviewing techniques used for navigation and guidance in closed environments, the authors evaluated and compared state-of-the-art in-house navigation solutions, and the implications of the research provide a summary of critical observations, some ideas, and guidelines for further development. It is essential to increase the perception of the target audience, increasing the acceptance of these devices.

Jin et al. [39], in their review, discussed the combinations of image processing techniques to create systems to map and locate indoor environments. The authors focused on solutions aimed at people with partial and total visual impairment to indicate the difficulties in providing fast and reliable responses. As a secondary theme of the work, the authors described that, in addition to the navigation process, which uses maps, other challenges require attention from researchers, such as the detection of obstacles present in the users’ routes. Computer vision models that manipulate two-dimensional (2D) and three-dimensional (3D) images are considered in terms of perceiving barriers (horizontal and vertical obstacles) and concerning processing consumption and information delivery time.

For this review, it was decided to provide details on technological and technical resources, describe the advantages and disadvantages of each one, and simplify and direct the reading, resulting in higher reliability of the information. Also, some classification schemes proposed in previous analyses are revisited to include information about the robustness of the concepts. For example, we considered Singh’s study, which classified network-based and non-network-based IPS systems and initiated a discussion on hybrid models using schemes based on magnetic sensors [38]. Here, we include other technologies, such as magnetic fields, ambient light, and audible or non-audible sounds, supplied or not in an encoded format, fingerprints, and the most advanced current technology, computer vision. We were also concerned about indicating multisensory arrangements and algorithmic combinations in hybrid models. Each technological and technical approach is described in detail below.

## 3. Indoor Positioning Systems

The technologies used in current positioning systems can be organized into two main categories: outdoor and indoor. In outdoor environments, several navigation systems are already well established and widely used, providing precise location and navigation services with error margins getting smaller and closer to zero [40]. These include Global Positioning System (GPS), Global Navigation Satellite System (GLONASS), and BeiDou Navigation Satellite System (BDS). However, in the indoor environment, the accuracy of satellite-based positioning decreases markedly due to signal losses when colliding with building structures, which causes the effects of multiple paths and delays in information delivery, thus not meeting the requirements of a reliable service location [41]. Indoor environments are where people spend about 80% of their lives and need to have reliable localization and navigation services for their occupants.

Traditionally, indoor positioning systems (IPSs) can be considered systems that work continuously and in real time to provide locations of humans in indoor areas [19]. IPSs can be used for many activities, for example, detecting and tracking people, assisting visually impaired people in their daily activities, and facilitating medical monitoring. Some public places, such as parks, museums, and shopping centers, need indoor positioning services, for example, to provide indoor navigation for visually impaired people, and to assist tourists and visitors in finding their destination [19,42].

In recent years, indoor positioning technologies and techniques have gained prominence, allowing the construction of systems for countless model scenarios [9,42,43]. A wide variety of technologies have been proposed, which is why IPSs can be categorized in several ways according to various criteria. For example, it is possible to classify them based on their architecture or on the way they determine the positions of indoor environments [11].

Indoor positioning systems classified based on their architecture can be divided into three classes: (1) self-positioning, (2) infrastructure positioning, and (3) assisted by self-directed infrastructure. In self-positioning architecture, devices determine their locations by themselves. In infrastructure positioning architecture, device positions are estimated using the locations of devices launched in the environment. In architecture assisted by self-directed infrastructure, an external system calculates the position and sends it to the tracked user in response to a request.

Indoor positioning systems classified based on the way they determine their most prevalent technology identifies the positions of indoor environments. IPSs mainly employ (1) radio frequency (RF), (2) inertial, (3) audible and non-audible sound, (4) light-based, and (5) vision-based technologies.

In this work, the indoor positioning systems were initially classified into two groups: based or not based on radio frequency signals. From these two classes, the systems follow an organization based on the technologies used to determine positions in indoor environments. Five subgroups were defined to indicate the predominant technologies used: radio frequency-based, inertial, sound, light-based, and computer vision-based systems. There is a sixth type, hybrid, which unites two or more technological approaches.

A complete classification of IPSs is shown in Figure 1. Hybrid systems are not represented in this figure, as there are many possible combinations of IPSs that can form a hybrid system, making a general classification unfeasible. In this work, we focus only on hybrid systems that can be scaled for navigation and indoor location applications for the visually impaired.

It is essential to know the performance and limitations of various options to select a suitable IPS, specifically for the visually impaired. Accuracy (i.e., position error) is usually a critical performance criterion for IPSs. However, other measures, such as response time, availability, and scalability, among others, also need to be considered. Considering that different applications require different types of IPSs in environments such as supermarkets and private homes, it is essential to know in detail each of the subcategories described in Figure 1. The next sections describe the systems, emphasizing their most typical applications and limitations.

### 3.1. Systems Based on Radio Frequency

Many different wireless networks can be deployed in an indoor environment, from traditional Wi-Fi networks, which are used in millions of buildings around the world to provide Internet access, to models with more specific purposes. Among the various models available are WSN models, which are designed to connect the Internet of Things (IoT) devices, popular RFID devices, and Bluetooth beacons, among other alternatives. A more simplified way to classify network-based indoor positioning systems is to consider how information is obtained. In this classification model, there are two groups: (i) methods based on range and (ii) methods without range.

Range-based methods extract geometric information (distance or angle) from signals from different wireless nodes and then combine the geometric constraints of each anchor to obtain the user’s position [43]. Free-range methods are based on the connectivity information between nodes or the identification of signal resource patterns that depend on location [1].

#### 3.1.1. Range-Based

There are different ways to extract geometric information from wireless signals. The most common are methods based on the signal propagation time between the transmitter and the receiver, the angle of arrival (AoA), or the intensity of the received signal (received signal strength indicator, RSSI) [44,45,46]. Below, we briefly describe the fundamentals of each class of methods, analyzing their advantages and disadvantages.

Time-based location: Time-based location uses algorithms that measure the propagation time of a signal between the transmitter and the receiver [44]. This location model is also known as the time of flight (ToF), and applies calculations that indicate the distance, d, between the user and the anchor node, as follows:(1)$$d=\frac{c}{4\pi f}\emptyset$$
where $c\approx 3\;\times \;10^{8}$ m/s represents the speed, and f is the signal’s modulation frequency. Commonly, 20 MHz are used, resulting in an unambiguous distance rage of 7.50 m. The most straightforward approach is known as the time of arrival (ToA). In ToA, the transmitter includes the time when the signal is transmitted in the radio packet, and the receiver calculates the time spent on the transmission. With this information, the receiver can calculate the distance between two points [9]. The user’s position can be obtained using a posteriorization method if the distance to several anchor nodes is known. The lateration method estimates the position as the point of intersection of three circles, as shown in Figure 2. The center of these circles is located at the position of the anchor nodes, and the radius is equivalent to the calculated distance estimate. In this model, three circles are needed to obtain a position estimate in a two-dimensional space.

The ToA method requires synchronization between all nodes, as the reference time must be the same in all cases [47]. This can be a problem for certain types of wireless networks with simple low-power devices and high restrictions on the complexity of the algorithm, such as WSNs. An alternative method that eases the limitation of synchronization between nodes is the time difference of arrival (TDoA). There are two main implementations of TDoA:Calculating the difference in ToA of a signal transmitted to two different receivers: For each TDoA measurement, the transmitter must be in a hyperboloid with a constant range difference between the two positions of the receiver [9]. This method relaxes the synchronization restriction for the receivers.Calculating the difference in ToA of two signals with different propagation times: Usually, the first signal is the radio packet, propagated in electromagnetic waves (speed of light, 300,000 km/s), and the second is a type of sound signal (340 m/s) [48]. This method eliminates the need for synchronization. However, all nodes must have additional hardware to send both types of signals simultaneously.

The use of time-based systems for indoor location has several limitations concerning its accuracy, caused by problems such as building structure, internal layout, and building location, which can cause the signals to be blocked [49,50]. To overcome this challenge, many studies present the application of combinations of techniques based on synchronized TDoA, based on an efficient hybrid synchronization protocol RBS-PBS. RBS (transmission synchronization in time required) and PBS (transmission synchronization in pairs) are protocols that indicate how the physical resources of the network are combined using codes that modify the number of nodes taking part in the arrangement. The use of these strategies makes the system more robust, reducing its margins of error when compared to the use of traditional TDoA [8].

More recent works are based on UWB technology, which improves the accuracy of the range due to the large bandwidth used [11]. The use of a large bandwidth allows implementing shorter pulses that increase the resolution of time and the accuracy of positioning estimates by the ToF method. The main problems of UWB technology, when applied to systems with a focus on large audiences, are the complexity and cost of implementation. Unlike Wi-Fi, UWB systems are not commonly found in indoor environments, and their deployment means an additional expense. However, this problem will be overcome with the implementation of the fifth generation (5G) of cellular networks worldwide, which will employ signals with high bandwidth and centimeter-level range accuracy [51].

Time-based location methods are susceptible to errors produced by clock inaccuracies, errors in time estimation, and a lack of synchronization between clocks [24]. For example, for a signal that travels at the speed of light, 1 µs of error corresponds to an approximate distance error of 300.00 m. Also, conditions with non-line of sight (NLOS) produce a positive bias in the distance estimate and should be included in methods of location and distance determination. The critical point of the approach is related to the increased complexity of the algorithms by including NLOS.

In general, time-based systems receive extremely accurate position estimates and can be applied over large areas due to their scalar characteristics. This scalability, however, does not guarantee users’ safety from collisions. Thus, an obstacle detection mechanism must be implemented, increasing the general complexity of the algorithm. Finally, the cost of the system is determined by the network technology and the number of nodes required.

Angle: Angle-based location methods adopt a signal’s angle of arrival as a criterion for calculating the position of the receiver [44]. Its operation is like time-based methods, but instead of using the distances to the anchor nodes, it checks the angles. There are usually two main methods of obtaining the AoA of a signal [52,53]:Use a matrix of sensors whose locations relative to the center of the node are known and use the difference in ToA of the signal at each sensor to calculate the AoA of the anchor node.Two or more directional antennas are pointed in different directions with overlapping main beams. The calculation of AoA takes place according to the RSSI proportion of the individual antennas.

The AoA value is estimated by calculating the triangulation of the signals received from one or more anchor nodes, as shown in Figure 3. If the position of the vertices of a triangle is known, it is possible to calculate the location of any node within the triangle, and thus know the angle at which the point sees the vertices [44,54].

In general, AoA systems are not widely used for the indoor location due to the need for additional hardware, such as sensor arrays or antennas [44]. Also, the calculation of angular estimates has high processing consumption. It is negatively affected by the low signal-to-noise ratio (SNR) and errors in the RSSI or ToA estimates [47]. These factors limit the scalability of the system due to the increase in monetary cost and consumption of computational resources.

RSSI: Location methods based on RSSI estimate the distance between the user and an anchor node using the strength of the received signal [42,55]. These methods consider that the attenuation suffered by a signal that travels from a transmitter to a receiver depends on the distance covered [56]. Thus, to estimate the distance between two points, it is necessary to model the environment with Wi-Fi devices using a signal propagation model. Traditionally, the log-distance trajectory loss model is used, where the attenuation (in dB) is equivalent to the logarithm of the distance covered [53], that is:(2)$$RSSI=P_{1m}-10\alpha \log_{10}d-\gamma \;$$
where d is the distance between the receiver and the transmitter, $P_{1m}$ is a reference power value measured in dBm at a distance of 1.00 m from the transmitter, and *a* is the exponent of the loss of path $\gamma \;\sim \;N\;\left(0,\;\sigma_{\gamma }^{2}\right)$ that models the effects produced by the shadow. Note that to obtain the model parameters, that is, $P_{1m}$ and α, it is necessary to perform a calibration. The calibration process consists of collecting the RSSI at predefined positions, with known distances to the anchor nodes, which is usually done using regression methods.

Once calibrated, the distance is estimated according to the path loss model using the maximum likelihood estimator (MLE), which for the case of the distance estimate is [46]:(3)$$d=10^{\frac{RSSI-P_{1m}}{10\alpha }}\;$$

As in the case of time-based location algorithms, the user’s position is estimated by combining distance information from various anchor nodes using a lateration method [45]. RSSI-based methods are attractive because of their simplicity, as RSSI measures are supported natively by most transceivers. Unfortunately, the variability of the Wi-Fi channel, together with the signal attenuation caused by building structures such as walls, or other elements such as the human body itself, introduces errors into the distance estimation and makes RSSI-based location algorithms less accurate than time- or angle-based algorithms [46].

An example of an RSSI-based localization system can be found in the work of Poulose et al., where the authors use correlations between RSSI samples in nearby locations to keep the user on a predefined path, adapting the value read to the propagation changes between areas of the same building [45]. Zanella et al. presented another example of using WSN, applying a cooperative method to locate nodes in an indoor environment [46].

In several studies, the authors use multiple receivers to improve the accuracy of the model, which, in its raw state, varies in RSSI measurements and provides approximate location estimates for the user [42]. Two of the approaches to indicate the efficiency in recovering a pre-defined location or path are the Cramér–Rao limit calculation in Gaussian and log-normal models [57].

RSSI-based systems can be easily scaled for large areas and multiple users due to the simplicity of their signal measurements. A feature that facilitates the application of the model based on RSSI is that measurements can be obtained without the need to be part of the network. Besides, the cost of the system’s infrastructure is minimal if existing Wi-Fi networks are used. Unfortunately, the accuracy provided by RSSI-based systems is low and is negatively affected by the distance between the transmitter and the receiver.

#### 3.1.2. Range-Free

Range-free methods are based on information obtained from the connectivity of a wireless network [25]. The values obtained can be used to estimate the position without calculating any measure of reach at an anchor node [58,59]. There are mainly two types of flawless algorithms: proximity and digital printing methods.

Proximity methods: These methods use the connectivity information to directly infer the position of the user based on the number of anchors in the neighborhood.

Proximity algorithms are based on a simple idea: if a user is receiving a signal from an anchor node, the user’s position must be close to the anchor node’s position [59]. The operation takes place according to the following steps: first, the user searches the channel looking for radio signals from the anchor nodes. After detecting an anchor node, the user’s position is estimated to the position of this anchor node. When there are several nodes available, the model will select the one with the strongest signal.

Figure 4 shows how the method works: the circles represent the coverage area of the anchor nodes, and it is necessary to create an intersection between them. When a user is around the circle maintained by node s1, the estimation of their position will use anchor node S1 as a reference. When a user is in the circle of anchor S2, the position will be estimated as the position of anchor node S2. The selection of the anchor node will be made in terms of the RSSI at the intersection of the two circles.

One example of using the proximity method is the Active Badge system [60]. This system uses a network of infrared sensors that detect signals transmitted by active tags and provides an accurate location estimate of a room. More recently, many authors have tried using Bluetooth Low-Energy (BLE) beacons to identify and locate targets.

Bluetooth is a wireless communication technology that uses information that is digitally incorporated into radio frequency signals [58,61]. The standard was initially intended for the exchange of data over short distances, with operating policies and protocol defined by the IEEE 802.15.1 standard [58]. The main objectives of the technology are to communicate between fixed and/or mobile devices to reduce the necessary infrastructure, eliminating cables and physical connectors, and to facilitate the synchronization of data and people [61].

Bluetooth technology has been considered as a resource for indoor position systems and a competitor to Wi-Fi. The widespread adoption of BLE occurred due to its availability (most modern smartphones support it), low cost, and extremely low power consumption. BLE devices allow fixed emitters to run on batteries for several months or even years [58]. Lin was among the authors who studied the use of BLE beacons in an indoor environment to track patients in a hospital [62]. Many authors have indicated that BLE technologies could be deployed in millions of buildings to provide the locations of places, objects, and people, without the need to invest in infrastructure, as indicated by Choi et al. and Fujihara et al. [63,64].

The level of error presented by proximity methods is causally related to the size of the coverage area. That is, if the coverage area is large, the error is increased, and if the coverage area is smaller, the number of anchor nodes required for the total coverage of an internal area increases. For this reason, several authors have experimented with applying complementary technologies, such as radio frequency identification (RFID), which allows reducing the level of error, with a low impact on the cost of the system [65]. Also, the use of passive RFID tags reduces the cost of maintaining the network, as the battery of the anchor nodes does not need to be replaced regularly.

A more general way of using connectivity information is to use the centroid algorithm, where the position estimate is calculated as the centroid of the position of the received anchor nodes [1], that is:(4)$$\hat{m}=\left(\frac{1}{N}\;\overset{N}{\underset{i=1}{\sum }}x_{i},\;\frac{1}{N}\;\overset{N}{\underset{i=1}{\sum }}y_{i}\right),$$
where $\hat{m}$ is the estimated position of the mobile node and $x_{i}$ and $y_{i}$ are the *x* and *y* coordinates of the *i*_th_ anchor node, respectively. In the centroid method, the accuracy of the position estimate depends on the number of nodes used in the calculation, and a margin of error that remains within predefined limits is tolerable. In general, the level of precision obtained is related to the order of the average distance between anchors implanted in the building. The advantage of the method is its computational simplicity, regardless of the physical or organizational aspect of the scenario.

Examples of RFID-based proximity methods can be found in the works of Singh et al., Qian et al., and Wu et al., where the authors review the RFID-based location methods available in the literature [66,67,68]. Proximity methods provide accurate position estimations of rooms using low-complexity algorithms [65]. The system’s scalability can be achieved by adding more nodes to the network to increase the coverage area. The cost of the system is low if passive RFID tags are used.

ZigBee is a wireless communication standard developed by the ZigBee Alliance, which was registered with the IEEE 802.15.4 standard as its physical layer and medium access control [69]. The model was proposed to meet the need for wireless networks with low energy consumption costs and data transmission rates [43]. The architecture of an indoor positioning system based on the ZigBee standard comprises a network of sensors that provide signals to algorithms already known and applied over a wireless sensor network [43]. These algorithms use RSSI values to estimate location, as in other Wi-Fi–Bluetooth techniques, and even in more sophisticated models, such as digital printing and signal propagation [70,71].

Fang built an indoor positioning system with ZigBee technology [72]. The author combined different approaches, such as signal propagation and fingerprinting, and calculated the best forecast for the location. The results of the experiments show that combinations of techniques provided better positioning compared to individual methods, including approaches using gradients and approximations of linear squares. The model’s precision level was around 1.25 m, which corresponds to 98.67% correctness on the mapped locations.

Fingerprinting methods: The fingerprinting location system uses the characteristics of information perceived in certain locations, which must be previously identified. Thus, the method requires two phases to be observed: an offline phase and an online phase. In the first phase, the area is divided into small cells to specialize in the identification of each place of interest, allowing the collection of data to form a database. The central idea of the digital printing method is to store a signal pattern in a similar way to a relational database. This pattern is used as a reference in the search for a value by equivalence or approximation. In the second phase, the position is estimated by comparing the captured data with the records stored in the database.

When the fingerprinting method is used to map locations through radio signals, the main concern is to perceive the singularity of the signals in different positions due to propagation problems caused by the complexity of indoor environments [2,27]. Usually, in indoor environments, different types of radio signals can be received, such as signals from Wi-Fi, WSN, or Bluetooth networks implanted in buildings, or coming from the external environment, such as GSM or LTE signals. The complexity of the indoor environment produces significant differences between the signals received at different locations due to multiple paths, shading, or propagation in NLOS environments [73,74].

There are two types of fingerprint methods: those that adopt deterministic strategies and those that adopt probabilistic strategies [19,75,76].

Deterministic models indicate high levels of accuracy, provided by high data consumption, the strong density of sensory coverage per square meter, or the combination of techniques to reduce spurious data, as perceived in SNR applications for RSSI signals on a Wi-Fi network. In addition to the k-nearest neighbor (KNN) and the weighted k-nearest neighbor (WKNN), other systems are also used to determine positions, such as support factors (SVMs) or linear discriminant analysis. Still, they have a high computational cost when compared to the KNN class algorithms [75,76].

In the group of probabilistic approaches, the objective is to find the location with the maximum probability of a set of candidate data. An example of the application of this approach is the Horus system, which uses the distribution of signals in the environment and calculates positions with the maximum subsequent probability [77]. Besides, there are also systems based on Bayesian networks and the Kullback–Leibler divergence [78]. The advantage of using fingerprinting with approximation methods is the reduction of computational and operational complexity since it is unnecessary to arrive at the exact value of the mapped record. The disadvantage of the probabilistic model is the complexity of scaling the system to larger areas.

The collection of fingerprinting is not limited to measuring the characteristics of radio signals. Recent works have proven that it is possible to use other types of information, such as the magnetic field [79]. The main advantage of fingerprinting methods used with magnetic field information for RSSI signals is that the magnetic field is more stable. While there are some approaches to indoor location using artificially generated magnetic fields, most modern systems use the strength and/or natural orientation of the Earth’s magnetic field to identify locations [80].

An IPS based on magnetic fields uses a magnetometer to measure variations in the magnetic field, which will be used to determine a person’s position [81]. The position estimation is commonly performed using methods such as digital printing. However, magnetic fields are less able to individualize the identification of a site, as the same value can be found in many different parts of a building unit. Typically, jobs and applications that use magnetic field-based fingerprints compare fingerprint strings instead of making point-to-point comparisons [82].

The sequences of magnetic fields are recorded in databases or adjacency matrices to improve understanding of the location [81]. Another action that helps to improve accuracy is to create an extensive collection of fingerprints for each point of interest. This process is time-consuming and vulnerable to environmental changes or the movement of humans or objects.

The accuracy of a fingerprinting method is causally related to the number of points raised from the mapping [55]. The size of the map can be an additional concern, as deterministic and probabilistic approaches affect its construction. Deterministic methods generally involve less calibration effort than probabilistic methods. In fact, finding the probabilistic values includes calculating the signal statistics at each calibration point [83]. Likewise, locating a person who is moving in an indoor environment requires a larger calibration scheme than locating a static object. Recently, several researchers have shown a keen interest in reducing the effort of the calibration process [84]. One way is to collect data about places using a collaborative approach, where several users contribute to the map as they use the system [14,84].

Regardless of the origin of the fingerprints, the main disadvantage of the method is the effort required to collect samples to compose the database, which increases the cost and reduces the scalability.

### 3.2. Systems Based on Inertial Sensors

Network-based systems estimate user positions by measuring the resources of signals received from a wireless network. This model requires an intervention in the environment with the implementation of physical infrastructure. Systems based on inertial information calculate their positions without having to apply external resources.

Estimating a future position given an initial position associated with speed and direction is one of the oldest methods of navigation, generally called “dead reckoning” [85]. An obvious problem with this model of dead reckoning is the increase in the progressive accumulation of errors since a small error of direction can mean a huge error over the distance covered [86].

Information about the position of a target must be accurate. Therefore, these models adopt the geographic coordinate model, composed of latitude, longitude, and altitude, to identify the absolute (inertial) position. The sensors allow the absolute (inertial) position to be immediately perceived in the roll, pitch, yaw (RPY) system using the x-, y-, and z-axes, based on the RPY (Figure 5) [87]. In such a system, the x-axis indicates the nominal direction (front), y (slope) is orthogonal to the x-axis and points to the left side, and the z-axis (yaw) points up [88].

Generally, inertial sensors are assembled, forming units of inertial measurement (IMUs), which have an accelerometer, a gyroscope, and a magnetometer, each with three axes [89]. The magnetometer is not an inertial sensor; however, in this work, it will be treated in conjunction with the other sensors because it is part of the IMU. The magnetometer sensor uses the earth’s natural magnetic field or artificial magnetic fields of alternating current to establish an address. Another sensor described in this study is the barometer, which indicates changes in the floor due to changes in atmospheric pressure.

Accelerometers can be used to determine changes in the user’s position when acceleration in each direction is detected. This is a very rough estimate, which can be improved using a gyroscope to observe changes in a course [7,90,91]. The evidence of initial acceleration can be confirmed by the fact that a user is walking, recognizing (also with an accelerometer) the typical tremor associated with walking [86]. In general, the pedestrian dead reckoning (PDR) algorithm behaves, as shown in Figure 6.

Dabove and Leppakoski reported the construction of inertial IPSs. They showed different strategies to reduce the accumulation of errors typical of inertial systems, refining the navigation with particle filtering and Kalman filter [88,91]. The system reported by Leppakoski et al. deals with a navigation map in which the user’s position is tracked continuously [88]. The filtering algorithms refine the general location by discarding impossible paths within the limits of the map, reducing the immediate errors, and accumulated errors over time.

Location and inertial navigation systems can be classified into two types [10]: strapdown systems and step and heading systems (SHSs). Strapdown systems use a person’s acceleration to estimate position. SHSs estimate the person’s location by the direction the head is pointing (direction) and by the length of the stride the person applies.

Regardless of the approach used, the first step in an inertial navigation system is to calculate the relative orientation of the sensor and the user’s body. IMU measurements are expressed in the sensor coordinate table and, depending on where the device is used, the user’s body. They may not correspond to the axes of the navigation frame.

Any misalignment in the axes produces errors of location and direction, necessary to estimate the relative orientation to keep the user’s displacement within a tolerable error field. The relative position of the user can be obtained by calculating the difference between the frame of reference coordinates and the values obtained from reading in each of the three axes (*x, y, z*), expressed in Euler angles, that is, roll (*x*), pitch (*y*), and yaw (*z*).

Strapdown systems: Strapdown tracking and inertial navigation systems are based on the concept that the double integration of acceleration obtains the user’s position. Thus, the first integration of the acceleration signal $a\;\left(t\right)=\left[a_{x}\;\left(t\right),\;a_{y}\;\left(t\right)\right]$ produces speed and the integration of speed provides position [92], that is:(5)$$v\left(t\right)=v\left(0\right)+\int_{0}^{t}\left(a\left(t\right)-g\right)\;dt$$
(6)$$m\left(t\right)=m\left(0\right)+\int_{0}^{t}v\left(t\right)\;dt$$
where v is speed, g is gravity, and m is position, all of which are related to the navigation frame.

Errors in sensor measurements affect position estimation in different ways. Errors coming from the accelerometer produce a drift in the position because the integration accumulates the errors over time. Errors from the gyroscope sensor result in an incorrect rotation matrix. In general, errors in estimating the position increase cubically with time due to the integration of the accelerometer and gyroscope signals [92].

Figure 7 shows a block diagram of a navigation system based on the strapdown model. In the strapdown model, initially, the angular speed measured by the gyroscope is used to indicate the orientation of the sensor for the navigation frame. Once the initial guidance is known, updated values can be obtained at any time. The orientation information is used as a reference to rotate the navigation frame, and the gravitation force is subtracted before integrating the acceleration signal to obtain the speed and position.

More recent studies have demonstrated that the use of a foot-mounted IMU reduces the time-dependence of position estimation errors in strapdown inertial systems. Foxlin et al. showed that location error, which usually grows cubically over time, is reduced to linear growth when the zero-velocity update (ZUPT) is used [93]. The idea of the ZUPT algorithm is to register the moment when the user’s foot is fully planted on the ground, considering that at this moment, the travel velocity is zero. Absolute zero is always not achieved, so the authors applied an extended Kalman filter to estimate errors in inertial measurement. The model showed deficiencies in indicating the position on an inclined floor, minimized with the application of a gyroscope, which updates the zero angular rate or applies a reduction in the heuristic direction [45,94].

Step and Heading Systems: Unlike strapdown navigation systems, step and heading systems do not use acceleration signal integration to calculate the user’s position [9,92,95]. Instead, these systems detect the user’s steps and estimate the length and direction of each step from the accelerometer and gyroscope signals. Then, the model recursively predicts the user’s position, accumulating vectors that represent the user’s movement at each stage, that is:(7)$$m_{x}\;\left(k\right)=m_{x}\;\left(k-1\right)+l_{step}\;\left(k\right)\;\cos\left(\theta \;\left(k\right)\right)$$
(8)$$m_{y}\;\left(k\right)=m_{y}\;\left(k-1\right)+l_{step}\;\left(k\right)\;\sin\left(\theta \;\left(k\right)\right)$$
where $m_{x}$ and $m_{y}$ are, respectively, the *x, y* components of the position, *k* is the time index, and $l_{step}$ is the duration of the step and q the position. The fundamental cycle for a system of steps and directions is as follows [10]:Identify the subset of data of an individual step.Estimate the step length.Estimate the heading.

Usually, detecting a pedestrian’s step is a task divided into two phases: (i) the support phase, when the foot is firmly planted on the ground, and (ii) the swing phase, when the foot is in the air. Most algorithms designed to identify stage events detect feet in the support phase on the ground, where the strategy is to monitor the lack of activity measured by the IMU (acceleration magnitude) during the support phase [10]. As an alternative, some methods detect repetitive movements to determine the moments when the feet are either on the ground or suspended [10].

Finally, the last point in the fundamental cycle of an inertial system is the estimation of the direction of the person’s head at each step. The direction estimation is obtained by integrating the gyroscope signal, which deviates by the errors accumulated during integration [88]. Fortunately, the error growth of the step and heading systems is linear over time, instead of cubic, as in the strapdown systems.

The user’s location is obtained by a magnetometer, which in indoor environments is influenced by ferromagnetic materials, and may result in a degraded position estimate [74]. The fusion scheme of data obtained from both the magnetometer and the gyroscope shows relatively good accuracy, with the deficiencies of each acting complementarily. That is, the gyroscope produces high-precision measurements in the short term, and the magnetometer provides low-precision measurements that are stable over time [74]. Kang describes an example of a step and direction system, where the authors designed a system for portable smartphones [96]. Likewise, Munoz built a system to be used on a smartphone resting in a user’s pocket to detect movement using the gyroscope signal [87].

Despite the improvements of inertial systems in reducing drift, it is still present. Therefore, this type of system cannot be applied for a long time without a correction strategy. Inertial systems are fully scalable in terms of size and number of users, as there is no need to deploy infrastructure in the environment. Furthermore, smartphones nowadays already include embedded IMU. Therefore, the cost of the system is minimal. However, as previously described, it is essential to consider the reduced precision produced by inertial drift.

### 3.3. Systems Based on Sound

Sound signals, consisting of pressure waves that propagate in the air, benefit from the fact that sound travels at a much slower speed than electromagnetic signals, thus allowing us to measure the time between the emission and arrival of a signal with much less effort [25,97].

The emission time is usually measured by the simultaneous transmission of a radio signal and an audible signal because the radio signal reaches the sensor almost instantly, and the sound signal reaches the sensor later. Therefore, the difference between these two times can be used to calculate the distance between two points. Sound-based systems can be classified into audible and inaudible sound systems.

Although audible and inaudible acoustic applications have shown excellent levels of accuracy, strong variations are also reported in certain scenarios, as the speed of sound is affected by temperature and humidity [50].

#### 3.3.1. Audible Sound

It is possible to use audible beeps to encode information for location systems. The idea of making an artificial audible sound has many disadvantages, especially considering that sound signals tend to irritate humans closer to the sound source [4]. However, other, more sophisticated schemes allow the inclusion of encodings that act as watermarks in the sound. The sound can be from a beep to a song played in a shopping mall or any other public place, with this encoding inserted in the audio being undetectable by the human ear.

For example, Li and co-authors presented an implementation of a user location system using digital audio signal watermarks, where a pseudo-random sequence was used to amplify several frequency bands of the host signal (which was music only) [98]. This model allows the insertion of different pseudo-random sequences to follow different targets in an indoor space. The authors claim that they obtained an accuracy of 1.30 m.

#### 3.3.2. Inaudible Sound

Sound signals considered inaudible to the human ear are emitted in bands below or above the audibility threshold. The human ear can perceive sounds between 20 and 20,000 Hz. Signals with a frequency below 20 Hz are called infrasound, and those with a frequency above 20,000 Hz are called ultrasound.

Indoor infrasound location systems are not quite common for several reasons, such as the cost and complexity of devices that work at this frequency. For example, Kenji explored the infrasound model to authenticate the identification of users in indoor environments by speech [25]. The authors described the difficulty in finding devices that work at this frequency since most available solutions are in the human audible range or above (ultrasound). For this reason, the authors described a way to obtain ultra-low frequencies of standard recordings generated by smartphones through acceleration-based cepstral features. To validate the experiment, the authors mapped more than 30 locations on a fingerprinting map. The level of accuracy of the system was over 90% in identifying the assigned areas.

Ultrasound location systems are quite common in automation approaches and have several hardware options available at a low cost. An evident advantage of ultrasonic signals over audible sound signals is that they are not detectable by humans [97]. Kang-Wook et al. used ultrasonic signals to determine the location of targets in indoor environments [97]. This study indicated that the model is limited by the need to establish a direct line of sight between the sensor and the monitored target. Even so, the results obtained in the experiments indicated a maximum error of 1.00 m only for the ultrasonic transducer and 0.35 m when the ultrasonic sensor was associated with a TDoA model, which explored the ultrasonic reflections of the sites. Brena led a study group on the use of ultrasonic sound and infrasound as tools to enable the identification of positions of places and people [4]. The experiments indicated an accuracy level of almost 0.01 m in the direct line of sight between the sensor and the monitored target.

### 3.4. Systems Based on Light

Although optical/light signals are only a form of electromagnetic radiation, we separate them from radio waves because the specific technologies are different, as well as their advantages and challenges. Thus, in this review, we consider light in terms of its visibility. Devices that emit frequencies outside the human visual field can be infra or ultra. In both approaches, the equipment or software deals with the luminous information in its completeness (complete light frequency) or in one of its channels [99].

#### 3.4.1. Visible Light

The transmission of data by visible light uses any type of lamp for this purpose; however, LED lights are the most suitable [5]. Visible light communication (VLC) uses the ability to turn the light source on and off at noticeably short intervals [100]. This on/off operation can occur so quickly that it is invisible to the human eye. The VLC for IPS has been tested in some indoor location studies, as described by Kim and a group of collaborators, who reused the artificial light infrastructure available in the environment to insert encodings [101]. This approach allowed the authors to reduce the cost of implementing their indoor positioning system significantly.

The principle of VLC is that each fixed lamp has different scintillation coding so that the sensor receives the light and compares the modulation with known coding schemes and associates it with the location close to the corresponding lamp [102]. An advantage of this arrangement is that it is non-invasive because human users see only ordinary lamps fixed in standard locations such as the ceiling. The receiver can be a photodiode or an optoelectronic device capable of capturing light intensity (for example, a photocell) or an image sensor (for example, a camera) to register the transmitter’s light pulses. The advantage of an image sensor is that it can record several lamps with their positions simultaneously, thus achieving more accurate location estimates.

All VLC-type projects are of the passive type, as the central element for the emission of information is the lamps, which are often heavy and need a connection to the power grid. Still, authors such as Guo and Zhang report accuracy levels below 0.20 m [5,99].

Some authors have experimented with the use of VLC in the Li-Fi mode [103]. The Li-Fi and LIDAR (Light Detection and Ranging) technologies, in addition to allowing the operation of guiding people in indoor environments, also reduce energy consumption, since transmission occurs only when users are detected in the environments. Another interesting feature of Li-Fi is its speed, which is much higher than that available in Wi-Fi technologies.

Most VLC systems locate the target based on information acquired from various luminaires (usually three or more), requiring a unique identifier, and the combined signal received from all visible luminaires needs to be separated [104]. The consequence of using several VLC devices is the need to apply multiplexing/de-multiplexing schemes.

#### 3.4.2. Nonvisible Light

The best known and most used approach is infrared (IR) [60]. A simple infrared system consists of a light-emitting diode, which emits infrared signals as bursts of nonvisible light, and a receiver photodiode to detect and capture light pulses, which are then processed to retrieve information [105]. The reliability of the IR system is affected by many characteristics of the emitted optical signal, e.g., its directivity or to what degree it is unidirectional. Another important aspect is the way it reacts to obstacles, such as reflection and dispersion (irregularities in reaching obstacles). Most infrared systems require distance from the emitter’s line of sight (LOS) to the sensor, although sometimes the reflected signals have enough energy to activate the sensor. Obviously, in the context of IR IPS systems, the lack of a direct view between the sender and receiver is a significant disadvantage, as the signal suffers from occlusion between the two devices [5].

A system developed by Wu et al. estimated the position of a target using a combination of RSS measurements in data networks, either fixed in the scenario or mobile, in robots [106]. The system arranged together with the user issues a unique infrared code every 10 s. The infrared sensors, which are placed around the office environment, capture the emitted codes. The information received by the sensor network is processed and made available on portable devices. The system has two limitations: it requires LOS between the receivers and the emblem, and the performance of the system is affected by sunlight. This system has been reported to compromise user privacy. During implementation, some users answered a questionnaire and were extremely uncomfortable to know that their locations were shared with other users [106].

Garrote et al. developed a study that aimed to estimate the location of a mobile robot due to its active configuration [107]. To assess the position of the target, the authors observed the measured distances between the destination and the predetermined reference points, using a system of non-linear equations of hyperbolic trilateration. The accuracy noted by the authors during the expressions was about 0.10 m.

### 3.5. Systems Based on Computer Vision

Indoor positioning systems (IPSs) that use computer vision use the information collected by cameras and a set of image processing techniques to identify, track, and navigate people in indoor environments [108,109]. Initially, we identified two classes of IPSs based on computer vision: fixed cameras, arranged in the infrastructure, and mobile cameras, available in wearable devices, smartphones, etc.

Another way of describing IPS is by observing the use of the cameras and the types of targets (passive or active). In structural systems, cameras are used actively, locating, and monitoring people in indoor environments. That is, the user does not need to load any device. Mobile systems can also be passive, in which the camera is used by the user to be monitored and captures images or videos from that user’s perspective. The captured information can be compared with previously stored files or learned models (decision trees, neural networks, etc.). Although the terms “active” and “passive” are not common in computer vision terminology, we maintain them to standardize the approaches to other technologies. In this review, we consider the terms “structure” and “mobile” to differentiate active and passive strategies.

#### 3.5.1. Cameras Fixed to the Scene

Shahjalal et al. introduced an indoor positioning system that uses a set of fixed IP cameras to reduce the rate of location errors [110]. In systems that use computer vision, the position is considered a primary issue to guarantee the confidentiality of the information generated. An application built for the Android platform receives images obtained from several simultaneous transmission links. Experimentally, the authors used a set of four cameras to provide the location of a target, resulting in a 0.10 m error margin.

Rao et al. built a computer vision project with an active approach (with a fixed camera), to make the environment intelligent in tracking people [111]. Users interact with the system by voice commands, and the system responds with audible instructions. The precision achieved was about 0.10 m in the horizontal plane. The weakness of the model is that it does not identify elements in the vertical plane. This deficiency (horizontal–vertical perception) is corrected with the use of stereo cameras, such as Microsoft Kinect [112].

#### 3.5.2. Mobile Cameras

Zhao et al. described that indoor location approaches that use a single type of sensory perception, under certain physical and logical conditions and dispositions, are more susceptible to failure [113]. One way to reduce the inaccuracies or limitations of these approaches is the combination of different sensory perceptions, composing a hybrid arrangement, which despite having a higher cost of implementation, provides more accurate and reliable results. The authors designed an indoor positioning system that combined the camera sensors, inertial sensors, and the Wi-Fi sensor available on smartphones to experience the levels of accuracy and limitations. The authors’ strategy was to combine stereo vision with data obtained from inertial sensors and Wi-Fi.

Some studies that use the passive mobile camera approach used visual odometry (VO) to update the user’s position. VO is the process of estimating the movement of an agent (for example, vehicle, human, or robot) using images captured by one or several cameras as input information. VO is not a recent approach, but it has been gradually improved in efficiency and accuracy [109]. An example of VO is presented by Xiaochuan et al., where a sequential set of stereo images was subjected to filtering to extract location information, keeping the consumption of computational resources low (low use of processes and memory) [6]. Another approach to indoor navigation adopts the structural perception of the scenario and the possible presence of obstacles through stereo vision [114,115].

### 3.6. Hybrid Indoor Positioning Systems

A hybrid positioning system, by definition, combines two or more systems to improve the performance of each individual system [116]. In a hybrid system, one of the technologies is generally considered more relevant for estimating the user’s location, while the others are considered complementary and are used to improve the system’s resources, such as accuracy and coverage area [43]. In some cases, technologies and techniques are fused to address the limitations of one of the components used in the positioning system. Currently, numerous hybrid positioning systems in the literature combine the different IPSs reviewed so far. A complete classification of hybrid positioning systems is not feasible due to the large number of possible combinations of IPSs that can form such systems.

IPSs that use a single type of sensor or algorithmic approach have certain limitations in the indoor environment due to their various characteristics, requiring the application of more physical or processing resources and time for data manipulation. For example, methods based on angle and time require additional hardware, influencing the cost of the system. Also, they cannot be applied to wireless networks if they are not synchronized with the network nodes. Fingerprinting methods require extensive data collection and calibration of results in all indoor environments.

In this section, we will review the most exciting hybrid systems for locating and navigating visually impaired people. This review focuses on the following groups: RSSI-IMU, RSSI-Vision, IMU-Vision, RSSI-IMU-Vision, and others. The mapping models adopted to improve the performance of the IPS will be described for each of them. Additionally, we will comment on the possibility of using smartphones as sensors and computational devices.

#### 3.6.1. RSSI-IMU Hybrid Systems

Many authors have experimented with the combination of Wi-Fi and inertial location. Studies have searched for better arrangements between the two models so that one will reduce the limitations of the other. A weakness reported in these studies is that pure inertial systems are weak because errors in estimation accumulate progressively, and pure Wi-Fi location systems are not accurate and responsive to user movements. Hybrid RSSI-IMU systems include methods that combine RSSI measurements with inertial measurements, using either a propagation model or a fingerprinting approach [95].

The availability of Wi-Fi networks deployed in millions of buildings worldwide makes RSSI-based positioning systems an attractive hybrid option because there is no need to invest in wireless infrastructure. One of the countless advantages of using Wi-Fi networks is the ability to calculate RSSI just by listening to the network, as most wireless communication standards already include the RSSI field in radio packages.

A hybrid strategy associates Wi-Fi signals to magnetic fields through the application of a particle filter and a Kalman filter [117]. In these studies, the objective was to create a collaborative scheme in which one technology compensated for the deficiency of the other [117]. Thus, due to the differences in Wi-Fi signals being greater than those perceived in magnetic fields, but providing a physical reference, some authors use Wi-Fi to indicate an area, and internally to this area, inertial information reinforces the perception of a location.

Zou et al. described an example of this type of hybrid system, in which the combination of visible light communication (VLC) with the inertial measurement unit (IMU) estimates the user’s position on the scenario map [118]. The authors applied a linearization method to determine quaternary and velocity errors with adaptive Kalman filter (AKF). An indoor positioning algorithm using the unweighted Kalman filter was developed to obtain the final positioning solution. The results show that the proposed integrated VLC and IMU algorithm reduced the errors of the positioning solution to less than 0.23 m. Shen et al. also combined the inertial system with RFID tags, which provide RSSI values, via Extended Kalman filter (EKF) [119]. Li, Poulose, and Correa described other examples of hybrid systems [9,45,120]. These authors configured the IMU sensor at the level of the user’s foot or hip and combined the inertial information with the position estimates of an RSSI-based system.

A concern in many studies is how to reduce the oscillations of signals obtained by inertial sensors caused by the environment itself and signals from other sensors [85]. Several authors have applied different strategies to improve the accuracy of the measurement to smooth out the noise with the use of filters. Akiyama and Zhao applied particle filters and Monte Carlo filters on inertial data to stabilize the values to be submitted to the location algorithm [59,85]. The authors reported an accuracy of about 1.50 m for the mapped reference place, and this value was much lower than the approximately 6.00 m using only fingerprints on Wi-Fi signals.

Poulose et al. and Correa et al. also presented a system that combines Wi-Fi and inertial location technologies [9,45]. Wi-Fi location is estimated using fingerprinting from signal strength measurements, and the inertial part identifies the user’s displacement and direction of travel by using the accelerometer and gyroscope sensors. The authors justified the use of fused Wi-Fi and inertial systems as a strategy to reduce the limitations of each component. Location by Wi-Fi signals cannot identify users’ movements, while inertial systems accumulate errors in the estimation of position over time. The data were combined using the Monte Carlo method, which applies a particle filter to eliminate the most dispersed records [121]. The results obtained in the study indicate an accuracy of 1.53 m in a 40.00 m walk. As a comparison, the authors redid the study with Wi-Fi signals only, obtaining an accuracy of 5.73 m.

#### 3.6.2. RSSI-Vision Hybrid Systems

Hanchuan et al. built a hybrid system by combining visual information with RFID markers [20]. This research used computer vision with 3D images obtained by a depth camera to record locations and RFID tags to retrieve the paths and improve the route indication for the user. The authors indicated that the accuracy was 96.6% for the sites stored as a reference on a route.

Zhao et al. also used the combination of Wi-Fi signals and visual information obtained by smartphone cameras in their work [113]. The authors’ concern was to keep the system continuously available to the user, with accurate and quick results, and to reduce the cost of implementing their solutions. The authors used Wi-Fi technology to extract visual and distance information to identify specific points, such as visible tags. The experimental results showed that the margin of error was 8%, providing an average distance of 0.2 m to the reference sites.

#### 3.6.3. IMU-Vision Hybrid Systems

Cheng et al. presented a study on a hybrid navigation method that combines computer vision with a set of inertial sensors to offer greater precision and efficiency in locating targets indoors [114]. The visual information is formed by a combination of two cameras (stereovision), which allows the calculation of a depth map and the distance for markers (reference points) or obstacles, in addition to reducing the errors accumulated by the gyroscope sensor. Inertial information is provided by a set of gyroscope, accelerometer, and magnetometer sensors. The Kalman dual filter (DKF) algorithm combines the data, which also reduces the accumulated error of the magnetometer. The results indicate that the method reached an average accuracy location level of 0.5 m.

Vaidya et al. proposed the construction of an intelligent home automation system to detect and recognize its residents through visual facial identification and tracking by a set of inertial sensors, available on smartphones [122]. The application responsible for facial recognition was run on a low-cost server, built on the Raspberry pi platform, which applied a Haar-like decision tree to extract the biometrics and compare with a photo base. Interactions with users occurred by sending audio containing instructions on the user’s location within the home structure.

#### 3.6.4. RSSI-IMU-Vision Hybrid Systems

Llorca et al. combined different technologies to improve the locating of people in indoor environments [115]. The authors started by indicating a person’s position through a combination of radio frequency-based technologies such as Wi-Fi, RFID, and Bluetooth Low-Energy (BLE). The combination of technologies recognized one or more people in an open area of a building and provided an ID for everyone. Due to the complexity of the environments, other sensory models were added to the model. Inertial technologies, which include features such as a magnetometer, gyroscope, and accelerometer, allowed the system to know the direction taken by the user while traveling. The computer vision allowed the system to provide more detail when identifying possible destinations that the user could reach. The distribution of technologies was defined as follows: The scenario received the Wi-Fi network nodes and the fixed cameras of the closed-circuit TV system. The user received a device that contained an RFID tag and active BLE beacons. The built model had the advantage of being scalable for the number of users monitored, and the error margins did not exceed 1.20 m.

Hanchuan et al. identified that the existing solutions do not identify locations with sufficient speed to allow continuous interactions, especially in heterogeneous and dynamic environments [20]. The focus of their work was to establish a system that had as a characteristic rapid, implicit, non-intrusive, and ubiquitous recognition of users or places of interest in indoor navigation. The proposed system has hybrid features, combining a 3D computer vision system (stereovision), inertial sensors, and RFID beacons, which allows one to quickly recover the path in a displacement performed at the speed of human walking (about 1.1 m/s). The results described in this work indicate that the system was able to simultaneously identify four individuals with 96% accuracy and seven with 95% accuracy.

#### 3.6.5. Other Hybrid Indoor Positioning Systems

In addition to the traditional radio-based sensors (Wi-Fi, RFID, Bluetooth, etc.), inertial sensors (barometer, gyroscope, accelerometer, magnetometer), and fixed or mobile cameras used in computer vision, other sensors are used in experiments and combined to provide indoor locations.

Martin-Gorostiza et al. described a solution for fusing infrared signals and images obtained by cameras, with the results inserted into a covariance matrix. A fusion algorithm based on the Monte Carlo filter consumes the matrix data and estimates the locations in indoor environments [105]. In the study, the authors combined a set of five infrared detectors and a camera to identify the position of a target by hyperbolic trilateration. It is obtained from phase difference measurements with the infrared sensor and by homography with the camera. The results reached a precision of around 0.70 m without the presence of multiple paths.

Guo et al. proposed a localization technique by fusing several classifiers based on the strength of the received signal (RSS) of visible light, in which different intensity-modulated sinusoidal signals emitted by LEDs are captured by photodiodes placed at various points in the grid [5]. The authors first obtained RSS readings to compose a fingerprint for each site. The signals were subjected to multiple data fusion and machine learning algorithms. Two algorithms proposed by the authors, grid-independent least square and grid-dependent least square (GD-LS), combine the outputs of the classifiers. Experiments were performed to show that the probability of having an average square positioning error of less than 0.2 m by the GD-LS is 93.03%.

## 4. Comparison of Systems Discussed

Several scientific and industrial research groups have proposed indoor positioning systems using different technologies, covering many types of sensors and models and logical algorithms with quite different characteristics. This advance in knowledge and mastery of physical and logical resources is essential for the proposed location and navigation services to be compatible with the users’ needs.

One of the main objectives of using these resources is to allow human activities to be mapped, offering products and services that increase the quality of life for users [24,61]. The visually impaired make up one of the groups of people who wait for indoor positioning solutions, especially those that offer safe navigation guides. Indeed, such systems would be a great impact on the quality of their lives due to the better relationship with the environments and the independence from other people’s help [82].

In this section, we present the demands identified by a group of authors and their proposals of navigation systems. A comparison of the studies is made to facilitate the understanding of the benefits and limitations of each approach.

When carrying out an evaluation, it is essential to observe the information provided by the authors themselves with some reservations, as their results generally do not follow independent evaluations [26]. One of the several reasons for the divergence between the values reported by the authors and those obtained in an attempt to replicate experiments is that the authors established conditions aimed at their investigation (ideal conditions), allowing to achieve the greatest possible precision, which is not realistic.

### 4.1. Application of Techniques and Technologies in Visually Impaired Navigation

Doush et al. researched the adaptation of indoor navigation systems for visually impaired people in the context of the scenario [123]. The authors’ objective was to identify the limitations in the countless strategies for using sensory technologies and algorithms to minimize user interactions with the system and provide continuous services, and automated and intelligent recognition about monitoring the environment. The authors classified the solutions found according to their types of deficiencies, confirming the need to explore the topic further and offer more reliable solutions.

Other authors, such as Nagarajan et al. and Aziz et al., take another classificatory approach to navigation and indoor location systems to assist visually impaired people [61,71]. The authors indicate that many strategies can be grouped into two broad classes: the systems that use maps and the systems of free mapping. In models that use maps, the systems depend on the data obtained by the sensors to establish addresses that allow representing faithfully or approximately the environments, such as buildings, schools, and hospitals [124]. On the other hand, authors such as Plikynas et al., Presti et al., and Caraiman et al. indicate that indoor environments have characteristics that make it difficult to read addresses through sensors and suggest as a solution strategies to perceive the scenario itself, within a radius of sensory vision [24,52,124,125].

Liu et al. created a hybrid tracking system for the Android platform to measure Wi-Fi signals and inertial data in different indoor settings [126]. The choice of using the smartphone was because of the possibility of using its sensors and its high availability among the visually impaired public. The authors’ strategy was to use inertial localization to mitigate the inaccuracy of data localization results presented by wireless networks, increasing the system’s reliability. In some places, where the absence of the Wi-Fi signal was perceived, the inertial location system would assume the role of providing the approximate position. The average error observed by the system was 15 degrees, indicating 0.80 m. In noisy environments or with metallic elements, the average error rises to 30 degrees, showing a margin of error of 3.0 m. The results showed that the hybrid location is robust in environments with the low noise level and may not be as robust in environments with the presence of metallic components that affect the operation of the magnetometer and the Wi-Fi sensor. A limit must be applied so that the system recalibrates the sensor in these cases to reduce this effect.

Galioto et al. formed a research group with interest in developing navigation for people with visual impairments in indoor environments using inertial sensors and the camera available on smartphones [23]. The inertial sensors magnetometer, gyroscope, and accelerometer provided information about the magnetic location and the direction to be followed in navigation in a PDR algorithm. At the same time, the computer vision indicated positions for the recognition of visual elements, such as colored ribbons and painted lines on the floor. The system demonstrated good accuracy in the navigation of visually impaired users. However, it is strongly impacted by lighting fluctuations, mainly in open places and with the incidence of sunlight.

Caraiman et al. developed a project called Sound of Vision (SoV) to provide an indoor navigation guide for people with visual impairments [52]. SoV’s premises were to meet a set of requirements such as being wearable, performing user guide operations in real time, respecting usability issues, and having a low acquisition cost. The navigation algorithm reconstructs the scene in its 3D aspects using a stereo camera (two cameras) to generate a safety area around the user associated with inertial sensors. The inertial sensor evaluates the user’s head movement (direction, speed) to provide angle correction parameters for the computer vision algorithm. The device, mounted on the user’s head, made it possible to perceive objects distant from the user about 0.15 m, using images with a stereo resolution of 2 × 1280 × 720 pixels, obtained at 15 frames per second (fps).

Many authors have also chosen to combine computational resources to make indoor navigation and location solutions adaptable to the context of the scenario. Li received collaboration from a team of collaborators to develop an indoor navigation model that would approximate the models adopted in outdoor scenarios [127]. The authors named the IndoorWaze system, which recognized the context based on crowdsourcing, increasing the level of knowledge of the scenario, using data previously mapped with data generated and labeled for the first time, provided collaboratively by other users. They submitted the two sets of data to the same fingerprinting map to represent the floor plan of building structures. The high complexity and cost involved in preparing maps for large environments such as shopping malls, condominiums, and university fields motivated the authors to use data provided collaboratively by users. The authors indicate that the maps enriched with data provided by the users generated a high-fidelity floor plan, with average error margins of 12%.

Simões et al. formed a research group on the location and indoor navigation of people with visual impairments [128,129]. The group’s two main research objectives were: to combine techniques and technologies to create records of reference sites to be used in the development of navigation routes and to create safety perimeters around the user to avoid collisions with obstacles.

Another approach for indoor navigation that detects obstacles in the path was proposed by Simoes et al. by the combination of stereo vision with a sound musical language. They varied the musical notes A+ and C+ in three tones (octaves) between treble, medium, and bass in the horizontal and vertical planes, providing the user with a 360-degree sound sensation [128] (Figure 8). The main idea of the work was to create a context-sensitive visual obstacle detection approach to help people with visual impairments to navigate indoors. The obstacle perception model uses a resolution of 320 × 240 at 10 frames per second (fps). This resolution allows effective ranges of about 4.00 m horizontally and a view of 2.50 m vertically when the camera is positioned on the user’s head facing forward and tilted about 45 degrees down from the point of the user’s point of view. They conducted the experiments in indoor environments with different characteristics, such as straight, curves, obstacles, low, medium (height of the user’s waist), and high (height of the user’s head). The error margins in the horizontal plane were 0.33 m and in the vertical, 0.20 m. In this experience, 95% of users rated the experience as excellent or good, while 5% of users indicated that it was difficult to follow the sound instructions.

Figure 9 illustrates the spatial recognition of the area around the user, indicating the viability of the path described by the guide and the possible presence of obstacles.

In the navigation research that used records, the authors established a mapping model that allowed the inclusion, editing, and removal of new locations without the need to redo the entire arrangement. They adopted a technique called Linear Weighted Policy Learner (LWPL), which takes advantage of the characteristics of the Weighted Policy Learner (WPL) algorithm. Still, it introduces a linearization of its formulations to consume less time and processing [129]. Each record is the result of the fusion of data from Wi-Fi signals (RSSI), inertial sensors, and images (computer vision). For the navigation process, a RAndom SAmple Consensus (RANSAC) filter was used to reduce the data captured in sample sets, while maintaining the representative robustness of the raw data. The data fusion technique for navigation differs from that used in mapping by applying a weighted average that recognizes the context for using or not using a given sensor. These data are compared with the LWPL map and a graph created by the Dijkstra algorithm representing the route. The navigation itself is performed by combining the pedestrian dead reckoning (PDR) algorithm with the Kalman Filter to make the PDR iterative and eliminate the accumulated errors generated by the inertial sensors. Figure 10 shows the architecture of the indoor navigation system for the visually impaired built in this research.

The algorithms were embedded in a prototype in the shape of glasses built according to the specifications suggested by the visually impaired users themselves and tested by a group of 20 people with different degrees of disability, ranging from low vision to the total blind. The margin of error between the position informed by the system and the reference position was 0.108 m and an average deviation of 0.186 rad.

### 4.2. Comparative Discussion on Technologies and Techniques

Numerous parameters have been used to compare IPSs, such as accuracy, type of location (2D or 3D), method (for example, triangulation or fingerprinting), algorithm, signal measurement (AoA, ToA, TDoA, RSSI), coverage, cost, etc. Based on these parameters, we consider that the most important ones when designing an indoor positioning system are precision (commonly indicated as an error value), scalability, the number of resources used in the studies in question, and the coverage area. These parameters were chosen because they are used in most projects and allow us to identify the benefits of using certain types of technology to develop an indoor positioning system.

Accuracy refers to the difference between the estimated and current position, as indicated by the system. As this difference can change according to the conditions, it is a concept that works with statistics. It must be expressed in terms of parameters such as distance, margin of error, and percentage, such as “error less than one meter in 90% of cases,” although authors rarely express precision in these terms.

Regarding the scalability criterion, the number of targets that the system can locate and navigate simultaneously in an indoor environment is discussed, observing how much its performance is affected.

The resources identified as anchors in this review refer to the physical resources utilized in the installation and operation of an indoor positioning system. For this record, cases in which the resources were already available, making up the existing infrastructure (for example, lighting, ambient sound, terrestrial magnetic field, network nodes, etc.), were considered.

Coverage is the extent of the area in which the system can locate users. Although some technologies can offer extensive coverage in an ideal environment, when used indoors, their coverage can be limited by numerous environmental factors. An IPS can locate persons within a range of meters or even find them on different floor levels within a building.

Based on these parameters, we examined the scalability criterion, the number of physical resources used in the system, the coverage area or scope of the model tested, and the accuracy (described as the error value). These criteria were used because they were identified as those commonly used in other studies and allow us to indicate the gain in the adoption of one technology or approach over others in specific scenarios. The information for each system is shown in Table 1.

Based on the parameters used by the various authors studied in this review, we maintained the criteria of accuracy, coverage, and cost, because we identified that these parameters are commonly used to indicate the advantage in choosing one technology or approach over others. For each technology presented, we combine their described accuracy, coverage, and cost to provide a macro view of the study. The information for each system is shown in Table 2.

## 5. Conclusions

This work researched indoor positioning systems (IPSs), with a focus on those that can be applied in the tasks of locating and assisting with the indoor navigation of visually impaired people. A general classification of IPSs was carried out. Each selected group was reviewed, analyzing the advantages and disadvantages of the systems in terms of precision and scalability. Then, hybrid systems were evaluated with a focus on current solutions available in the literature. All solutions were also analyzed for their use on mobile devices, especially on smartphones, as these devices are popular worldwide and have the advantage of having most of the sensors used in IPSs, which reduces the cost of mass deployment of an IPS.

Considering the requirements in building applications for the visually impaired, which require high levels of accuracy, information delivery time, and scalability, it is possible to observe that in the literature, the approaches described as hybrid enrich the information generated by the algorithms. Hybrid arrangements can create more reliable or complementary information so that some data can cover the deficiencies of others in specific locations.

It is essential to point out that today, smartphones already have several built-in technologies, such as Wi-Fi, Bluetooth, BLE, IMU sensor, and an increasing number of high-definition cameras, which can be added to mapping and location schemes to increase system accuracy.

Current technologies have shown significant advances in terms of accuracy and speed and have been used in new algorithmic arrangements to improve the quality of internal positioning systems. However, the problem of positioning in indoor environments is far from being solved, and it needs further research to obtain the accuracy required by certain types of users, such as visually impaired people. Besides, in the case of indoor positioning systems based on technologies that use special-purpose devices and specialized infrastructure (for example, sensor networks, readers, and encoders), the cost of acquisition, installation, and maintenance is high. Transmitting this high-cost perception to the end-user, in many cases, makes its acceptance unfeasible.

There is still no satisfactory solution to the problem of an IPS capable of being used in all indoor scenarios with standard behavior. Fully accurate solutions are costly, or they are not in real time, or cheaper proposals are very inaccurate.

In practice, the evolution of technologies and algorithms has had an incredibly positive impact on the growth of indoor positioning systems. Changes in new technologies have given software designers greater freedom to experiment with new approaches that were previously nonviable due to their limitations. Also, new technologies offer the opportunity to close gaps in the research and can be used alone or combined with other, already tried models.

The details of the technology, technique, and implementation affect the accuracy of the system. For example, Faragher and Luo used internal positioning based on triangulation [58,133]. Faragher used Bluetooth and had an average error of 2.08 m, while Luo used VLC and indicated a much lower margin of error, with about 0.10 m in the tests.

Based on the comprehensive review of the literature, this research can encourage new research efforts, and for this reason, we suggest here some lines of future research in this area. Some criteria must be stringently met in future approaches, such as the ability to adapt to numerous positions of the user, the heterogeneity of transmitters and receivers, and the consumption of energy.

Adaptability to indoor positioning: The most adherent algorithms and technologies (acceptance by those who use them to build systems and by those who use the final product) are those that allow their application with a higher rate of adaptation. For example, portable systems can be offered for applications on smartphones and other devices that can be used in various positions on a person, such as in a pocket, purse, or the user’s hand, fixed at the height of the head, etc.

Heterogeneity: In an indoor location and an indoor navigation application for the visually impaired that is intended to be made available to large masses of users, the hardware and algorithms must be able to deal with the various characteristics of closed environments. Multiple wearable devices, such as smartwatches, smartphones, and the countless gadgets presented in recent years, can be used to provide different sensitivities and data patterns.

Power consumption: It is known that battery life on many commercial smartphones is not as good as desired. The same is true of other wearable devices. Therefore, it is crucial to develop new algorithms that are energy-efficient, with reduced computational complexity.

In several works described in this review, we note the efforts of the authors to solve problems of location in indoor environments. They deal with different characteristics that interfere with the signals and data received by the sensors and try to take advantage of the technologies and techniques already consolidated for the external environment. This means that, although current studies are looking for a model for indoor environments, in the future, the solutions will be more comprehensive and will be integrated with indoor or outdoor location services as they become available.

Other problems related to the global positioning models (external and indoor) that require attention are privacy and security. In this research, few authors have identified the need to preserve information about the user’s location during the experiment, such as Chopra et al. and Zheng et al. [13,134]. Other authors treated security as an essential item for users and observed that its application may or may not influence the decision to adopt a location system [9,81].

Although privacy has been a concern since the beginning of the development of IPSs, it is understandable that the matter now is to define the best algorithms and technologies in the future, and this will be one of the primary considerations for the adoption or choice of specific IPSs.

## Figures and Tables

**Figure 1 sensors-20-03935-f001:**
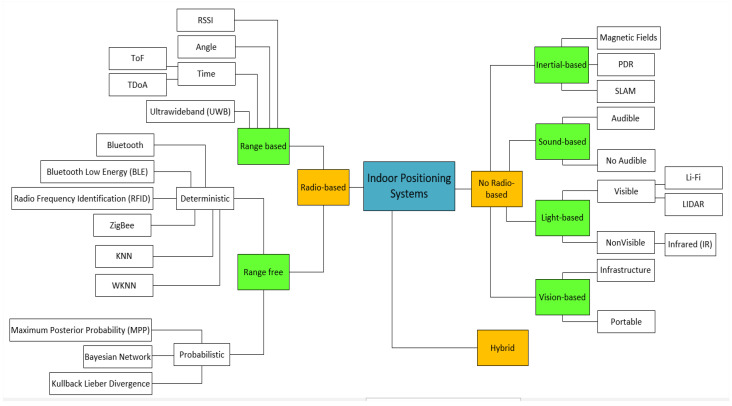
Classification of indoor positioning systems (IPSs).

**Figure 2 sensors-20-03935-f002:**
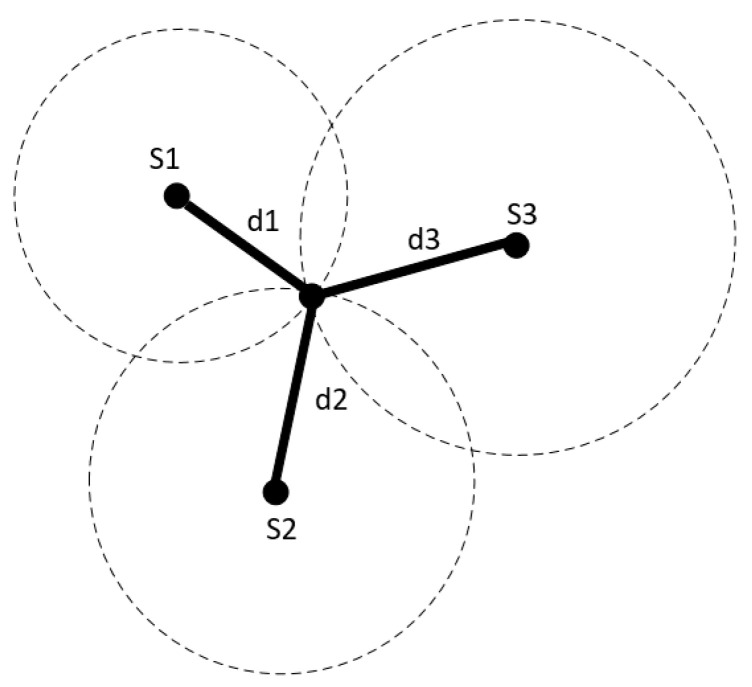
Lateration method between three nodes (S1, S2, S3) indicating their distances (d1, d2, and d3).

**Figure 3 sensors-20-03935-f003:**
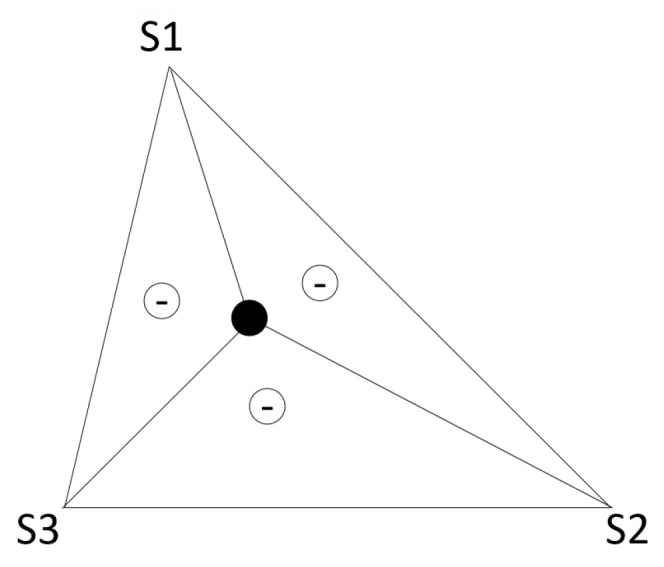
Method of trilateration between nodes S1, S2, S3.

**Figure 4 sensors-20-03935-f004:**
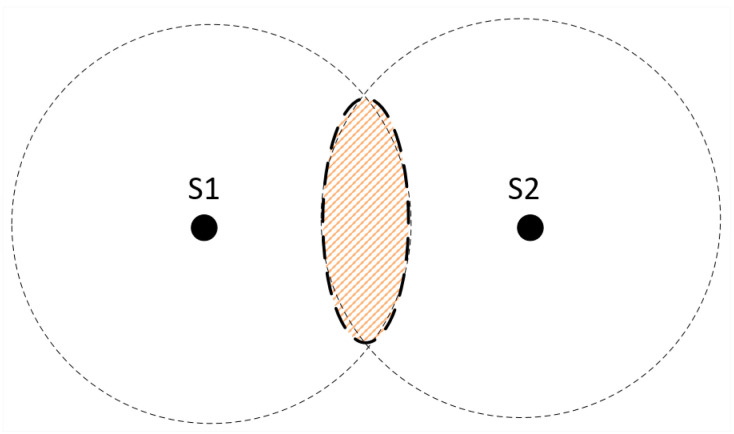
Proximity method concept.

**Figure 5 sensors-20-03935-f005:**
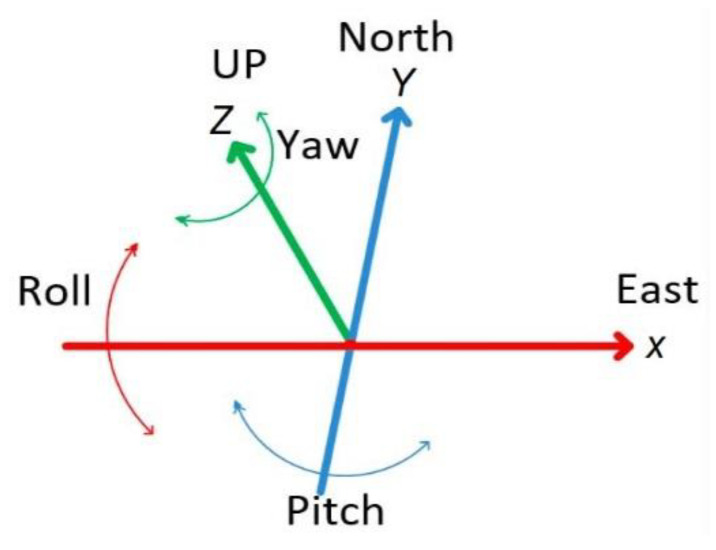
Roll, pitch, yaw (RPY) system reference.

**Figure 6 sensors-20-03935-f006:**
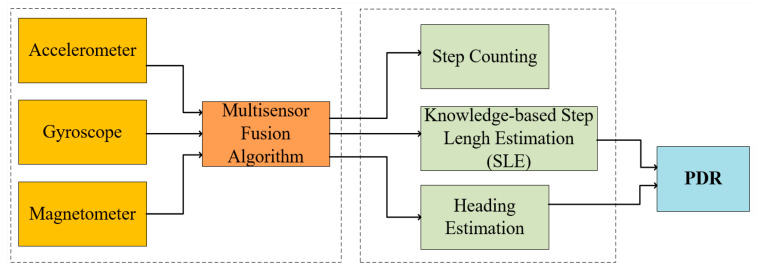
Pedestrian dead reckoning (PDR) scheme.

**Figure 7 sensors-20-03935-f007:**
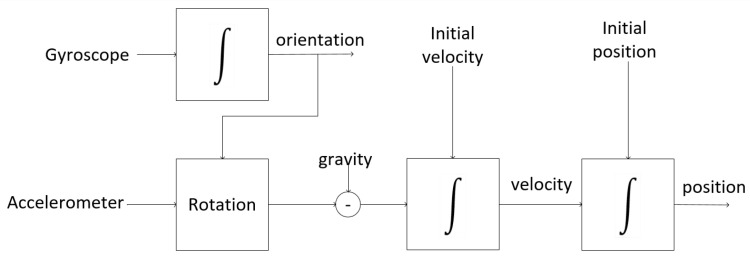
Strapdown navigation system.

**Figure 8 sensors-20-03935-f008:**
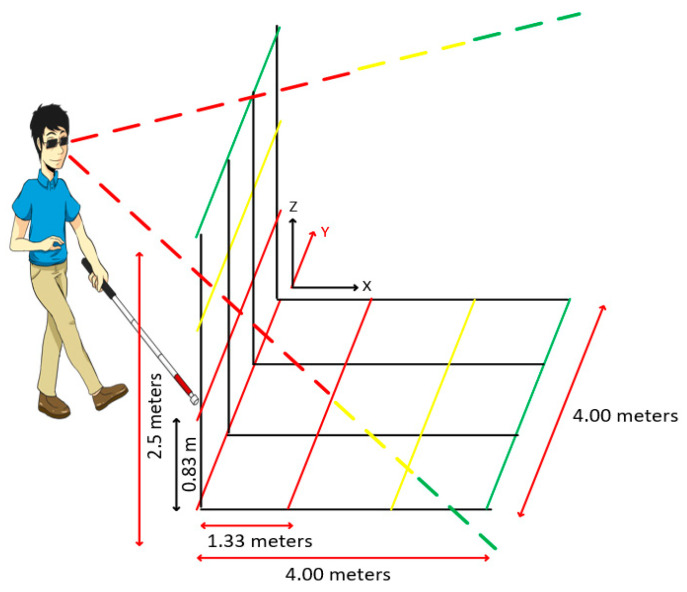
Obstacle detection scheme using stereo vision.

**Figure 9 sensors-20-03935-f009:**
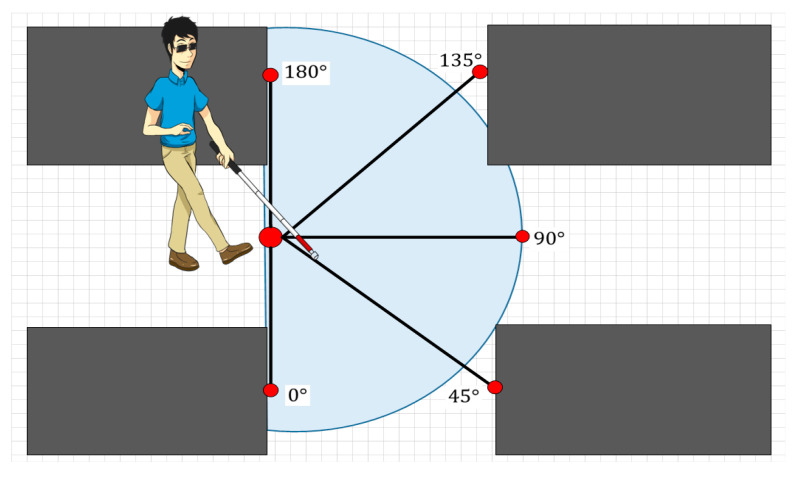
Spatial recognition of the area around the user.

**Figure 10 sensors-20-03935-f010:**
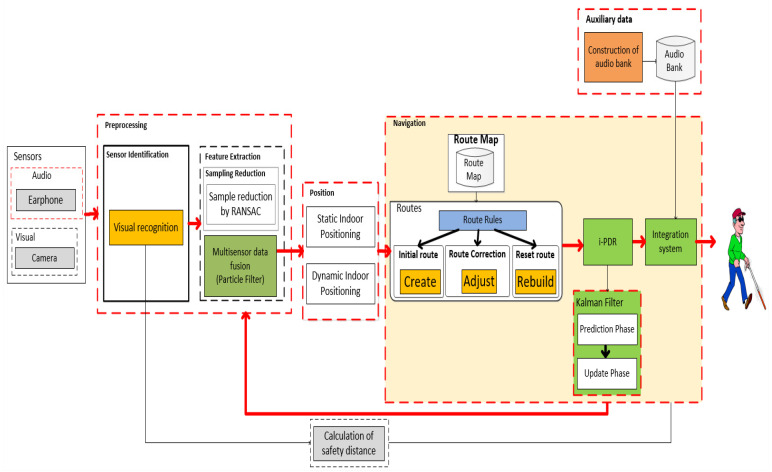
Indoor navigation system architecture.

**Table 1 sensors-20-03935-t001:** Overview of indoor positioning systems.

System	Type	Scalability	Limitations	Error Value
Hlaing et al. [49]	TDoA ^1^	Limited	Time	1.34 m
Zafari et al. [19]	RSSI ^2^/SNR ^3^	Yes	Does not identify the direction	2.40 m
Li et al. [98]	Audio	Limited	Noise	1.30 m
Kam-Wook et al. [4]	Ultrasonic/ultrasonic with TDoA ^1^	Limited	Line view	0.35–1.00 m
Guo et al. [5]	VLC ^4^		–	93.03% (0.20 m)
Shahjalal et al. [110]	Fixed IP camera	Limited	Requires high processing	0.10 m
Rao et al. [111]	Fixed IP camera	Limited	Work only horizontal plane	0.10 m
Zhao et al. [113]	Mobile camera/inertial sensor/Wi-Fi ^5^	Limited	Navigation direction identification	92% accuracy (0.2 m)
Zou et al. [118]	Adaptive Kalman filter, VLC ^4^/Wi-Fi ^5^/inertial sensor	Limited	Optical angles of incidence and irradiance should not exceed the field of view limitations	0.23 m
Garrote et al. [107]	VLC ^4^/hyperbolic trilateration	No	Optical angles of incidence and irradiance should not exceed the field of view limitations	1.10 m
Akiyama et al. [85]	Monte Carlo filter Inertial sensor	No	Processing time	1.50 m
Zhao et al. [59]	Particle filter Inertial sensor	No	Processing time	1.50 m 6.00 m (only Wi-Fi ^5^ signals)
Poulose et al. [45]	Wi-Fi ^5^/inertial sensor	Yes	Complexity in adding/removing network nodes	1.53 m 5.73 m (only Wi-Fi ^5^ signals)
Li et al. [20]	Camera/RFID ^7^	Yes	Distance	96.6%
Cheng et al. [114]	Kalman filter Inertial/camera (stereovision)	No	Insufficient acquisition of visual information during displacement	0.50 m
Llorca et al. [115]	Wi-Fi ^5^/RFID ^6^/BLE ^7^/inertial/camera	Yes	Distance	-
Li et al. [20]	Camera 3D/inertial sensors/RFID ^6^	No	High processing and network consumption, interference from other sources emitting infrared signals	96%
Martin et al. [105]	Infrared sensor/camera	No	High computational cost, interference from other sources emitting infrared signals	0.70 m
Hlaing et al. [49]	TDoA ^1^	Limited	Time	1.50 m
Gala et al. [130]	Wi-Fi ^5^	Limited	Requires additional infrastructure	3.0 m
Correa et al. [42]	Wi-Fi ^5^/inertial	Yes	Fluctuations in Wi-Fi ^5^ values and cumulative error of inertial sensors	1.4 m
Palumbo et al. [56]	RSSI ^2^	Yes	Distance	1.8 m
Lin et al. [62]	BLE ^7^/proximity	Yes	Requires additional infrastructure	97.22%
Bolic et al. [65]	RFID ^6^/proximity	Yes	Passive RFID ^6^ tags cannot perform complex operations, such as proximity detection and location	0.32 m
Zafari et al. [19]	Fingerprinting	Limited	High computational cost to add/remove records	2.0–69.0 m
Han et al. [131]	Fingerprinting	Limited	Room layout affects signal strength	3.0–9.0 m
Youssef et al. [77]	Fingerprinting	Limited	High level of complexity for tracking multiple targets	1.4 m
Kuang et al. [8]	Magnetic fingerprinting	Limited	Motion estimate error	2.5 m
Norrdine et al. [132]	Inertial	Yes	Cumulative error of inertial sensors	0.3–1.2 m
Teng et al. [12]	Inertial	Limited	Cumulative error of inertial sensors	1.0–2.0 m
Li et al. [120]	RSSI ^2^, inertial (SHS ^8^)	Yes	Requires additional infrastructure, has low accuracy, electromagnetic interference, low security, and long response	4.00 m
Shen et al. [119]	RSSI ^2^, inertial	Yes	Fluctuations in Wi-Fi ^5^ values and cumulative error of inertial sensors	1.35 m
Fang et al. [72]	ZigBee	No	Requires additional infrastructure	98.67% (1.25 m from the reference point)
Liu et al. [126]	RSSI ^2^, inertial	Limited	Requires additional infrastructure, has low accuracy, need to recalibrate	0.8–3.0 m
Galioto et al. [23]	Mobile camera, inertial	Limited	Cumulative error of inertial sensors, Optical angles of incidence and irradiance should not exceed the field of view limitations	92.01% (1.48 m from the reference point)
Caraiman et al. [52]	Kalman filter Inertial, camera (stereovision)	Limited	Cumulative error of inertial sensors	0.15 m
Simoes et al. [128]	Kalman filter camera (stereovision)	No	Lateral perception failure above 15 degrees	0.33 m (horizontal plane), 0.20 m (vertical plane)
Simoes et al. [129]	RSSI ^2^, inertial, Camera Kalman filter, Particle filter	Yes	High level of complexity for tracking multiple targets	0.108 m, 0.186 rad
Li et al. [127]	RSSI ^2^, Fingerprinting	Yes	Time, Requires additional infrastructure	88.0% (1.60 m from the reference point

^1^ TDoA—Time Difference of Arrival ^2^ RSSI—Received Signal Strength Indication ^3^ SNR—Signal-to-Noise Ratio ^4^ VLC—Visible Light Communications ^5^ Wi-Fi—Wireless Fidelity ^6^ RFID—Radio-Frequency Identification ^7^ BLE—Bluetooth Low-Energy ^8^ SHS—Step and Heading Systems.

**Table 2 sensors-20-03935-t002:** Comparison of leading indoor positioning technologies.

Technology	Precision	Weaknesses
TDoA ^1^	1.34–1.50 m	Infrastructure
RSSI ^2^	1.80–6.00 m	Low precision, access point
RSSI ^2^/ SNR ^3^	2.40 m	Infrastructure
RFID ^4^	0.32 m	Very low precision
ZigBee	0.25 m	Special equipment
Audio	1.30 m	Sensitive to audio noise
Ultrasonic	1.00 m	Infrastructure
Ultrasonic with TDoA ^1^	0.35 m	Infrastructure
VLC ^5^	0.20–0.23 m	Infrastructure
Fixed camera	0.10 m	Sensitive to light conditions
Mobile camera	0.20 m	Sensitive to light conditions
VLC ^5^/Wi-Fi ^6^/inertial sensor	0.23 m	Infrastructure
Inertial	0.30–2.50 m	Sensitive to the presence of metallic materials and people
Inertial with camera	0.50 m	Sensitive to light conditions
Wi-Fi with inertial	1.35–4.00 m	Sensitive to the presence of metallic materials, people, and blocking signals by the infrastructure
Wi-Fi ^6^ with camera	0.20 m	Sensitive to light conditions
Infrared with camera	0.70 m	Sunlight, sensitive to light conditions

^1^ TDoA—Time Difference of Arrival ^2^ RSSI—Received Signal Strength Indication ^3^ SNR—Signal-to-Noise Ratio ^4^ RFID—Radio-Frequency Identification ^5^ VLC—Visible Light Communication ^6^ Wi-Fi—Wireless Fidelity.

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
