# Peer review of "A Review of Technologies and Techniques for Indoor Navigation Systems for the Visually Impaired"

_sensors, 2020, doi:10.3390/s20143935_

Round 1

Reviewer 1 Report

The paper tackles the categorization of the complex world of the indoor positioning and navigation systems. Attempts in the same direction have been proposed many times and, in my opinion, this paper does not offer a step forward with respect to the state-of-art. The paper contains a good idea trying to specialize the content only to systems able to offer services to visually impaired users. But, this promising goal does not match with the content of the article.

The paper still readable and interesting but the problem of novelty is quite important. Furthermore, Figure 1 does not represent all the possibilities for ILSs, in fact, for example, the problem to be context-aware, or not, is not mentioned. I suggest staying focus on systems for visually impaired users, trying to propose some typical use cases and try to highlight which kind of techniques and technologies are promising with respect to a particular and very interesting scenario as it is. Otherwise, the discussion in section 4.1 is weak because too many heterogeneous techniques and technologies are mixed together without the possibility, for a reader, to understand why, for example, some technologies perform better than others. 

Author Response

A Review of Technologies and Techniques for Indoor
Navigation Systems for the Visually Impaired
Walter C. S. S. Simões, Guido S. Machado, André M. A. Sales, Mateus M. de Lucena, Nasser Jazdi, and Vicente F. de Lucena Jr.

In blue is the reviewer’s statements. Next, we present our proposed solution.

Issue #1: The paper tackles the categorization of the complex world of the indoor positioning and navigation systems. Attempts in the same direction have been proposed many times and, in my opinion, this paper does not offer a step forward with respect to the state-of-art. The paper contains a good idea trying to specialize the content only to systems able to offer services to visually impaired users. But, this promising goal does not match with the content of the article.

Answer:
We appreciate your review and comments to enrich the content of this manuscript. We understand that if the chosen approach were to indicate techniques and technologies simply, it would not be new and would not bring gains to the area. However, we realized in the various readings carried out that there is a multitude of organizations presenting the resources (physical and logical) for the construction of indoor navigation systems and that requires an enormous effort to know what their levels of accuracy, weaknesses, costs of processing, etc., is. That is why we created Tables 1 and 2. Anyway, we improved the manuscript covering the other reviewers’ suggestions and hope to have reached a quality level that allows the publication of this work.

Issue #2: The paper still readable and interesting but the problem of novelty is quite important. Furthermore, Figure 1 does not represent all the possibilities for ILSs, in fact, for example, the problem to be context-aware, or not, is not mentioned. I suggest staying focus on systems for visually impaired users, trying to propose some typical use cases and try to highlight which kind of techniques and technologies are promising with respect to a particular and very interesting scenario as it is. Otherwise, the discussion in section 4.1 is weak because too many heterogeneous techniques and technologies are mixed together without the possibility, for a reader, to understand why, for example, some technologies perform better than others.

Answer:
We appreciate your contribution to making our work more robust and complete. We accepted the suggestion and inserted new discussions on the use of context-aware techniques and technologies aimed at people with visual impairments. For example, on page 19, there is an indication of the use of computer vision as a resource for navigation that uses records and maps and navigation based on scenario perception, using stereo vision (references [118] and [120]). On page 3, we identified authors who were concerned with the relationship between accuracy and delivery time of information for visually impaired users, who are highly dependent on the logical and physical tools for safe navigation (references [20] and [21]). On page 5, we indicate that these concerns are aggravated when scenarios require a high level of accuracy, such as hospitals, requiring that algorithmic strategies be combined to detect and track partial and total visual impaired users in these contexts (references [19] and [42]). On page 19, we also describe hybrid arrangements that act collaboratively to keep the user within a monitoring field (reference [130]).
On page 23, a subsection called Application of Techniques and Technologies in the Navigation of the Visually Impaired was inserted, in which studies are described whose focus was to allow safe navigation of users (covering users with low vision to the total blind) in indoor environments.  References such as [132], [23], and [52] indicate the challenges of deciding to apply maps to assist in the elaboration of routes and navigation guidance or to create resources that maintain the physical integrity of the user, preventing collisions with obstacles. These studies show how the authors chose the combinations among the countless options based on the context of the scenario and the user’s profile.
Context-Aware-based systems are included in the hybrid studies in Figure 1, which deal with combinations of technologies and techniques to provide the ability to adapt systems to the context. To describe such hybrid characteristics whose focus is to provide updated information to users continuously, we have inserted a discussion in section 4.1. In this new section, we describe the
application of techniques and technologies aimed at meeting the location and navigation demands of visually impaired people, exploring the limits of each component, and forming combinations that complement each other taking into account the real-time scenario. We considered references such as [136], [62], and [72] whose presented approaches were context-aware based. We also included two studies carried out by our research team whose focus was to offer a robust navigation system adapted to the variations of the scenario. In both, we used a combination of sensory information [134] and independent navigation based on the information obtained from mapped records (that may modify following the environment dynamic), with the concern to keep the user safe from impacts with obstacles along their way [133].

Thank you very much for your kind review.
Best regards,
Walter Charles Sousa Seiffert Simões
Corresponding author

Reviewer 2 Report

This paper provided a review of technologies and techniques for Indoor navigation systems for the visually impaired. This topic is challenging but important. This paper summarized three main aspects of the relative researches, and most are clear written. However, some aspects need to be improved.

-the researches about indoor navigation by zone partition are not included.

-the structure of the 'related work' line 92 is not clear, it is like some pieces jointed together casually

-please check the eq.(1), and is it proper to leaving the comma in the equations? including all the equation below.

-It is not proper in the 'finger print' part line 472 not mention the important traditional WKNN method and its relative works

- This paper lacks the researches about the Kalman filter using in the indoor navigation which is known as the most basic and important technique.

Author Response

A Review of Technologies and Techniques for Indoor
Navigation Systems for the Visually Impaired
Walter C. S. S. Simões, Guido S. Machado, André M. A. Sales, Mateus M. de Lucena, Nasser Jazdi, and Vicente F. de Lucena Jr.

Second Reviewer:

In blue is the reviewer’s statements. Next, we present our proposed solution.

Issue #1: The researches about indoor navigation by zone partition are not included.

Answer:
We appreciate your suggestion to enrich the manuscript. We inserted a case study that applies the zone partition in section 4.1 to make this navigation strategy more evident. Among the cited references, some give light to models that apply zone partitioning, directing the strategy to the uses of sensors available on smartphones (references [36], [37], [125]).
We have already mentioned publications such as [36] and [24] that used schemes for combining zones by grouping sensors and map cells by clustering. In these works, the authors describe their localization approaches that combine mapped records, which have small fields of view in an enlarged field of view. This strategy of clustering records increases the ability to keep the user monitored in the environment and allows to activate only the sensors of the cluster, leaving the others in a waiting state, thus consuming less electricity. Other studies use zoning by
fingerprinting to create a zone for identifying the user’s location, such as the references [16] and [125]. Other studies use zone partition navigation to design a security perimeter around the user, as in [101] and [109].

Issue #2: The structure of the ‘related work’ line 92 is not clear, it is like some pieces jointed together casually

Answer:
We reviewed the paragraphs starting on line 92; red characters identify the modifications.

Issue #3: Please check the eq. (1), and is it proper to leaving the comma in the equations? including all the equation below.

Answer:
We checked equation 1, and we found an error (we used the wrong equation), thank you very much for your precise observation. We fixed it to indicate the relationship between speed and signal frequency. Regarding the use of commas, all equations without this error are now correct.

Issue #4: It is not proper in the ‘finger print’ part line 472 not mention the important traditional WKNN method and its relative works

Answer:
We appreciate your suggestion and included studies that address the WKNN model among the methods (page 3).

We hope to have answered your questions and improved the quality of the manuscript. Thank you very much for your comments and your thorough analysis of the text and concepts.

Best regards,
Walter Charles Sousa Seiffert Simões
Corresponding author

Reviewer 3 Report

see attached

Author Response

A Review of Technologies and Techniques for Indoor Navigation Systems for the Visually Impaired
Walter C. S. S. Simões, Guido S. Machado, André M. A. Sales, Mateus M. de Lucena, Nasser Jazdi, and Vicente F. de Lucena Jr.

Third Reviewer:
In blue is the reviewer’s statements. Next, we present our proposed solution.

Issue #1: The structure is generally quite good, however, it was somewhat confusing to mention magnetic systems from within the radio frequency section. I believe this deviation occurred because of the choice to describe, in detail, the various positioning techniques in the RF section. It is difficult when describing indoor positioning systems to separate the techniques used (AoA, TOA, RSS, etc,) from the technology used (RF, inertial, magnetic, light) as they are inherently linked concepts, but it can become repetitive to describe every technique for every technology. In this case, I think that it would to better to split magnetic systems into their own subsection.

Answer:
We appreciate your thorough review of our manuscript. We had some difficulty in isolating the techniques from the technologies, as many of them are common and serve as a standard for experimenting with new approaches. Thus, one way to minimize any redundancy of text about technologies and their uses in the most diverse strategies was to observe their context of origin. For example, AoA, TOA, was initially applied to approaches with technologies based on radio frequency and later used to technologies based on inertial information, light, computer vision, etc. Another issue that presents itself as a difficult factor to isolate are inertial systems and systems based on magnetic fields. Some references, such as [38] and [80], call the group inertial and insert magnetic systems as a member. Other studies, such as [3] and [10], indicate that magnetic systems do not make up inertial schemes and are treated separately. We chose to keep the inertial sensors (gyroscope, accelerometer) together with the barometer and magnetometer sensors in the same section because they are used in a combined manner in most approaches that apply models such as PDR. We have given greater emphasis to magnetic models, such as text on page 12, by references [81], [82], and [83].
We hope you accept this proposal, but we may change if you still believe our approach compromise the paper’s quality.

Issue #2: I think the information on visible light positioning (VLP) is light on detail and the survey would benefit from further information on this topic. In particular, because the motivation for this paper is navigation for people with visual impairments, VLP is a real contender due to accuracy being a significant strength of this technology.

Answer:
Your observation about the use of VLP is indeed important: the model presents numerous possibilities for its use in indoor navigation and location systems. Thus, we broadened the discussion on VLP by targeting studies that focused on the visually impaired due to their necessary characteristics of information accuracy. One of the references points out the advantages of using Li-Fi type VLP and compares its benefits concerning the Wi-Fi model [126]. Another study deals with the use of LIDAR to map and navigate indoor environments [127]

Issue #3: In the related work section, it may be worth discussing the recent meta-review on indoor positioning: G. M Mendoza-Silva, J. Torres-Sospedra, and J. Huerta, “““A Metareview of indoor positioning systems”” “, Sensors, vol. 19, no. 20, p. 4507, Jan. 2009, doi: 10.3390/s19204507.

Answer:
We are grateful for the suggestion of references that address indoor positioning approaches, especially those that shed light on the discussion about the evolution of the sensory algorithms and technologies adopted in the studies. We promptly received the suggestion of M MendozaSilva, J. Torres-Sospedra, and J. Huerta’s work and inserted it as one of our references, enriching
the discussion (reference [50]).

Issue #4: Also, in the related work section, this reviewer finds the attribution of collaborative works to single authors (and the possible assumption of male gender) to be quite strange. The use of ‘et al.''' and the removal of gendered pronouns would be far more inclusive and show recognition of the collaborative nature of science.

Answer:
Thank you very much for this observation about the manuscript rudeness. You are completely right. That was a stupid mistake of us, and please forgive. Research is always collaborative work, and there is no place for sexism or any other form of privilege to any group. Thus, we accept your suggestion to remove the references to a single author, redoing the paragraphs.

Issue #5: Finally, more discussion on the specific application mentioned in the title would be very helpful. Navigation for visually impaired people is a very important application of indoor positioning and well worth discussing. It also brings with it certain unique requirements. The authors identify precision and accuracy as very important in their discussion, but fail to mention that it is even more important for visually impaired people as they do not have the benefit of integrating the positioning system output with their own visual system. It would probably be most useful somewhere near the end when comparing and discussing the various systems to tie this all back to the application.

Answer:
We appreciate your contribution to making our manuscript more robust and complete. We accepted the suggestion and inserted new discussions on the use of techniques and technologies aimed at people with visual impairments. For example, on page 19, there is an indication of the use of computer vision as a resource for navigation that uses records and maps and navigation based on scenario perception, using stereo vision (references [118] and [120]). On page 3, we identified researches that were concerned with the relationship between accuracy and delivery time of information for visually impaired users, who are highly dependent on the logical and physical tools for safe navigation (references [20] and [21]). On page 5, we indicate that these concerns are aggravated when scenarios require a high level of accuracy, such as hospitals, requiring that algorithmic strategies be combined to detect and track partial and total visual impaired users in these contexts (references [19] and [42]).
On page 23, near the end of the manuscript, a subsection called Application of Techniques and Technologies in the Navigation of the Visually Impaired was inserted, in which studies are described whose focus was to allow safe navigation of users (ranging from low vision to the total blind) in indoor environments. References such as [132], [23], and [52] indicate the challenges of deciding to apply maps to assist in the elaboration of routes and navigation guidance or to create resources that maintain the physical integrity of the user, preventing the even collide with obstacles.
Our group has already published studies on the subjects of electronic mapping and indoor navigation for visually impaired people, where our objectives were to provide a system adaptable
to the context of the scenario, using the combination of several sensory models and many algorithmic strategies. Our most recent published works are “Audio Guide for the Visually Impaired People Based on Combination of Stereo Vision and Musical Tones” reference [133] and “A Hybrid Indoor Positioning System Using a Linear Weighted Policy Learner and Iterative PDR” reference [134]. The study using an audio guide is independent of scenario mapping and aims to guarantee a safety
perimeter around the user using stereo vision to calculate the distance to obstacles [133]. The study of hybrid navigation uses a mapping scheme with sample reduction to cover an indoor area
with records to be related by neighborhood levels in the routing and navigation algorithm [134]. The demands we faced on those researches inspired us in writing this manuscript aiming to help other researchers to expand even more this quite interesting topic.

Issue #6: On a side note, the page numbering restarts twice during the paper. This happens around the landscape oriented pages.

Answer:
Thank you for letting us know about this format issue. We fixed it in this new version.

Thank you very much for your kind review.
Best regards,
Walter Charles Sousa Seiffert Simões
Corresponding author

Round 2

Reviewer 1 Report

Authors have successfully answered to my previous concerns. In my opinion, the paper can be published. 

Author Response

Dear Mr. Reviewer,

We would like to thank you for all the contribution and suggestions for improvement given to our work. The revisions were made with care, seeking to comply with all comments that were relevant to arrive at this latest version.

Best regards,

Walter Charles Sousa Seiffert Simões